# Fast constrained sampling in pre-trained diffusion models

**Alexandros Graikos**
Stony Brook University,
Stony Brook, NY
agraikos@cs.stonybrook.edu

**Nebojsa Jojic**
Microsoft Research,
Redmond, WA
jojic@microsoft.com

**Dimitris Samaras**
Stony Brook University,
Stony Brook, NY
samaras@cs.stonybrook.edu

## Abstract

Large denoising diffusion models, such as Stable Diffusion, have been trained on billions of image-caption pairs to perform text-conditioned image generation. As a byproduct of this training, these models have acquired general knowledge about image statistics, which can be useful for other inference tasks. However, when confronted with sampling an image under new constraints, e.g. generating the missing parts of an image, using large pre-trained text-to-image diffusion models is inefficient and often unreliable. Previous approaches either utilized backpropagation through the denoiser network, making them significantly slower and more memory-demanding than simple text-to-image generation, or only enforced the constraint locally, failing to capture critical long-range correlations in the sampled image. In this work, we propose an algorithm that enables fast, high-quality generation under arbitrary constraints. We show that in denoising diffusion models, we can employ an approximation to Newton's optimization method that allows us to speed up inference and avoid the expensive backpropagation operations. Our approach produces results that rival or surpass the state-of-the-art training-free inference methods while requiring a fraction of the time. We demonstrate the effectiveness of our algorithm under both linear (inpainting, super-resolution) and non-linear (style-guided generation) constraints. An implementation is provided at this GitHub repository.

## 1   Introduction

The development of large text-to-image models [22, 27, 26, 30] has made denoising diffusion [32, 16] the go-to approach for capturing complex data distributions in high-dimensional spaces, such as images. By training on billions of text-image pairs, these models have acquired general knowledge about the image space, beyond text-to-image generation. This knowledge is useful in quickly adapting to new conditions [45, 40] and utilizing model features to solve downstream image tasks [38, 39].

The simplest way to utilize this prior knowledge is by fine-tuning the model. However, fine-tuning may require non-trivial computational resources and thus, previous works have focused on developing algorithms for conditional generation using *only* the pre-trained model [5, 28, 6, 44, 13]. These methods modify the diffusion sampling process by computing additional gradient terms that move the sample towards the condition while denoising. When these gradients are computed using backpropagation through the model weights, there is a significant increase in inference time. On

39th Conference on Neural Information Processing Systems (NeurIPS 2025).

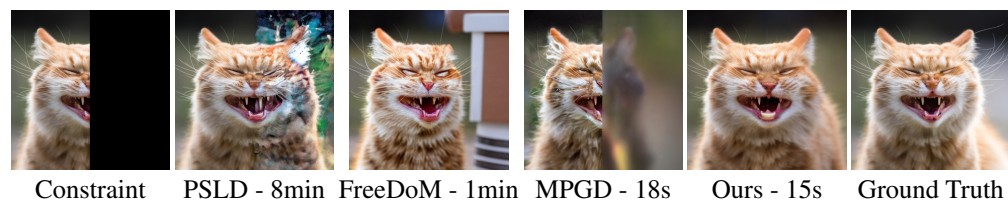

| Constraint | PSLD - 8min [28] | FreeDoM - 1min [44] | MPGD - 18s [13] | Ours - 15s | Ground Truth |

Figure 1: When tasked with completing the missing half of an image, previous methods are slow and fail to capture the important long-range dependencies between pixels. The proposed algorithm generates a reasonable image at a fraction of the time.

the other hand, when attempting to save computation by not propagating the condition information through the model, the generated image fails to capture the necessary long-range correlations.

As an example, in Figure 1, we use different algorithms to fill in the missing half of an image. Methods that backpropagate through the denoiser model (LDPS, PSLD [28], FreeDoM [44]) require significantly more time to run and do not consistently produce realistic results. Algorithms that do not compute gradients through the denoiser (MPGD [13]) fail to propagate the condition to distant pixels. With the shortcomings of existing approaches in mind, we pose the following question: Can we avoid backpropagation through the model weights while at the same time maintaining long-range consistency when updating the sample with the conditioning signal?

We view the denoiser network as a function that removes all noise from an input image. In that context, the goal of constrained sampling is to find how the input should change such that the clean output satisfies the condition better. This involves the denoising function's Jacobian matrix, which transforms a local change in the input to a change in outputs. We show that existing methods that backpropagate the constraint through the model employ the transpose of the Jacobian to invert this transformation, corresponding to the gradient descent direction.

An alternative direction is given from the Newton method [24], but requires more computation as it involves finding the Jacobian inverse. However, the input and output of the denoiser are closely related; when the intensity of some pixels in the final, noise-free image is increased, the same pixels are also expected to brighten in the noisy image and vice versa. Therefore, we propose to approximate the Jacobian inverse in the Newton step with the Jacobian matrix itself. We show that this is a valid approximation that (a) is fundamentally different from the gradient descent direction used in previous works, (b) is cheap to compute, with only two forward passes through the denoiser required, and (c) can quickly converge to the desired solution in large pre-trained diffusion models.

To evaluate, we generate images under both linear and non-linear constraints. We first show that our approach matches the results of state-of-the-art methods on free-form inpainting and $8\times$ super-resolution at a fraction of the inference time. We then demonstrate how existing methods fail at inpainting large regions, while our algorithm obtains results closer to a fully fine-tuned diffusion model on inpainting. Finally, we show how we can apply our algorithm to non-linear constraints and perform style-guided and mask-guided generation, where the proposed method consistently generates images that satisfy the constraints better than existing approaches.

## 2 Background

### 2.1 Denoising Diffusion Models

Denoising diffusion models [32, 16] have been widely adopted due to their exceptional ability to synthesize diverse and high-quality samples. The original formulation treats the training and inference process as a hierarchical latent variable model $x_T \to x_{T-1} \to \cdots \to x_1 \to x_0$, where the final latent is distributed normally $x_T \sim N(\mathbf{0}, \mathbf{I})$ and $p(x_0)$ represents the data distribution. Given a noise schedule $\alpha_t$ that defines the forward transitions $x_t \to x_{t+1}$ that corrupt the data with Gaussian noise, usually centered at $\sqrt{\frac{\alpha_{t+1}}{\alpha_t}} x_t$ and with variance $(1 - \frac{\alpha_{t+1}}{\alpha_t})$, the model is trained to reverse each step in the diffusion process by predicting the noise added to the clean sample. The predicted noise is shown to approximate the score function $\nabla_{x_t} \log p_t(x_t)$ [34] of the diffusion latent variables.

Further iterations of denoising diffusion introduced classifier guidance [9], which adapted a pre-trained unconditional diffusion model for conditional sampling by training an additional classifier $p(\boldsymbol{y}|\boldsymbol{x}_t)$. During inference, each reverse step also includes the gradient $\nabla_{\boldsymbol{x}_t} \log p(\boldsymbol{y}|\boldsymbol{x}_t)$, which guides the diffusion latent $\boldsymbol{x}_t$ towards regions that also satisfy the condition $\boldsymbol{y}$. However, if the conditioning $\boldsymbol{y}$ is already given, training this additional classifier on top of the diffusion model is costly. Classifier-free guidance [15] eliminated the need for an additional classifier by incorporating the conditional guidance into the base diffusion model training process.

The most widely adopted formulation of denoising diffusion is Latent diffusion models (LDMs) [27]. LDMs reduce the complexity by modeling the compressed latent space of an autoencoder instead of images. With an encoder $\mathcal{E}$ and decoder $\mathcal{D}$ that accurately reconstruct images $\boldsymbol{x}_0 \approx \mathcal{D}(\mathcal{E}(\boldsymbol{x}_0))$, the diffusion process can be made more efficient as redundant information in an image is left for the decoder to reconstruct. Most large text-to-image diffusion models, such as Stable Diffusion [27, 23], are based on the LDM approach.

## 2.2 Gradient descent steps for constrained sampling

Previous works studied whether large pre-trained diffusion models, which required significant investment to train, can be directly used for inference under novel conditions without additional tuning for each different constraint [5]. The typical problem formulation is denoising the sequence $\boldsymbol{x}_T, \boldsymbol{x}_{T-1}, ..., \boldsymbol{x}_1, \boldsymbol{x}_0$ under a linear constraint on the final signal in the form $\boldsymbol{A}\boldsymbol{x}_0 = \boldsymbol{y}$, or a relaxed version that minimizes $||\boldsymbol{A}\boldsymbol{x}_0 - \boldsymbol{y}||_2^2$ as part of the likelihood $p(\boldsymbol{y}|\boldsymbol{x}_0) = \mathcal{N}(\boldsymbol{y}; \boldsymbol{A}\boldsymbol{x}_0, \sigma^2 \boldsymbol{I})$.

Contrary to classifier guidance, which trained a separate model for the likelihood $p(\boldsymbol{y} \mid \boldsymbol{x}_t)$, the linear constraint only applies to the final, noise-free image $\boldsymbol{x}_0$. Thus, existing methods rely on Tweedie's formula [10], by which denoising diffusion models approximating $\nabla_{\boldsymbol{x}_t} \log p_t(\boldsymbol{x}_t)$ can be used to express the expected value of $\boldsymbol{x}_0$, denoted as $\hat{\boldsymbol{x}}_0$. Including the constraint as if an additional observed variable $\boldsymbol{y}$ was generated requires adding $\nabla_{\boldsymbol{x}_t} \log p(\boldsymbol{y} \mid \boldsymbol{x}_t)$ to every diffusion sampling step, and previous works considered different approximations of $p(\boldsymbol{y} \mid \boldsymbol{x}_t)$ using the estimated $\hat{\boldsymbol{x}}_0$ [5, 44].

Regardless of how the constraint gradient is applied, the regular denoising diffusion steps are altered so that at each $t$, the generated latent $\boldsymbol{x}_t$ is moved in the direction reducing the cost

$$C(\boldsymbol{x}_t) = (\boldsymbol{A}\hat{\boldsymbol{x}}_0(\boldsymbol{x}_t) - \boldsymbol{y})^T (\boldsymbol{A}\hat{\boldsymbol{x}}_0(\boldsymbol{x}_t) - \boldsymbol{y}). \tag{1}$$

For example, in the case of inpainting missing pixels, the matrix $\boldsymbol{A}$ extracts the subsection of the known pixels in image $\hat{\boldsymbol{x}}_0$ to be compared with a given target $\boldsymbol{y}$. The estimated expected value of $\boldsymbol{x}_0$ at the end of the chain is provided by the diffusion model as a nonlinear function $\hat{\boldsymbol{x}}_0(\boldsymbol{x}_t)$ learned by the denoiser network during training. Typically, these moves are gradient descent moves, i.e. moves of $\boldsymbol{x}_t$ in the direction of $-\boldsymbol{e}_t$ where

$$\boldsymbol{e}_t = \nabla_{\boldsymbol{x}_t} C(\boldsymbol{x}_t) = \boldsymbol{J}^T \boldsymbol{A}^T (\boldsymbol{A}\hat{\boldsymbol{x}}_0 - \boldsymbol{y}) = \boldsymbol{J}^T \boldsymbol{e}, \tag{2}$$
$$\boldsymbol{J} = \nabla_{\boldsymbol{x}_t} \hat{\boldsymbol{x}}_0(\boldsymbol{x}_t), \quad \boldsymbol{e} = \boldsymbol{A}^T (\boldsymbol{A}\hat{\boldsymbol{x}}_0 - \boldsymbol{y}).$$

Note that we name the error signal $\boldsymbol{e}$ as the (negative) direction of the gradient w.r.t. $\boldsymbol{x}_0$ itself, and the $\boldsymbol{e}_t$ is the matching move in the noisy $\boldsymbol{x}_t$.

The matrix $\boldsymbol{A}^T$ inverses the operation of $\boldsymbol{A}$ and can usually be computed for each task a-priori. Since computing the full Jacobian $\boldsymbol{J}$ is impractical, the gradient is instead computed using backpropagation through $C(\boldsymbol{x}_t)$, which yields the direction $-\boldsymbol{J}^T \boldsymbol{e}$. Chung et al. [5], [6], for example, apply gradient steps of this form to $\boldsymbol{x}_t$ at each step $t$ of generation before moving on to the next stage. As optimization of $\boldsymbol{x}_t$ might reduce the total noise in the image below what the denoising at $t-1$ was trained for, the gradient steps moving $\boldsymbol{x}_t$ towards optimizing $C(\boldsymbol{x}_t)$ can be combined with adding additional noise, which could also be seen as a form of stochastic averaging, as done by Yu et al. [44].

## 3 Method: Approximate Newton steps

We start by observing that a Newton optimization step, instead of moving $\boldsymbol{x}_t$ in the gradient descent direction $\boldsymbol{J}^T \boldsymbol{e}$, moves it in the direction $\boldsymbol{J}^{-1} \boldsymbol{e}$. We demonstrate it on a more general form of the cost

$$C(\boldsymbol{x}_t) = (f(\hat{\boldsymbol{x}}_0(\boldsymbol{x}_t)) - \boldsymbol{y})^T (f(\hat{\boldsymbol{x}}_0(\boldsymbol{x}_t)) - \boldsymbol{y}), \tag{3}$$

for some target $\boldsymbol{y}$ to be matched with projection function $f$, which in the linear case is $f(\boldsymbol{x}) = \boldsymbol{A}\boldsymbol{x}$. The Newton optimization would first approximate

$$f(\hat{\boldsymbol{x}}_0(\boldsymbol{x}_t - \boldsymbol{e}_t)) \approx f(\hat{\boldsymbol{x}}_0(\boldsymbol{x}_t)) - \boldsymbol{J}_f \boldsymbol{J} \boldsymbol{e}_t, \tag{4}$$

where $\boldsymbol{J}_f$ is the Jacobian of $f$. We then rewrite the cost of move $-\boldsymbol{e}_t$ as

$$C(\boldsymbol{x}_t - \boldsymbol{e}_t) = (f(\hat{\boldsymbol{x}}_0(\boldsymbol{x}_t)) - \boldsymbol{J}_f \boldsymbol{J} \boldsymbol{e}_t - \boldsymbol{y})^T (f(\hat{\boldsymbol{x}}_0(\boldsymbol{x}_t)) - \boldsymbol{J}_f \boldsymbol{J} \boldsymbol{e}_t - \boldsymbol{y}). \tag{5}$$

The cost is minimized by setting $\nabla_{\boldsymbol{e}_t} C = 0$ to get the system

$$\boldsymbol{J}^T \boldsymbol{J}_f^T (f(\hat{\boldsymbol{x}}_0(\boldsymbol{x}_t)) - \boldsymbol{y}) = \boldsymbol{J}^T \boldsymbol{J}_f^T \boldsymbol{J}_f \boldsymbol{J} \boldsymbol{e}_t \Leftrightarrow \boldsymbol{e}_t = \boldsymbol{J}^{-1} \boldsymbol{J}_f^{-1} (f(\hat{\boldsymbol{x}}_0(\boldsymbol{x}_t)) - \boldsymbol{y}) \tag{6}$$

$$\boldsymbol{e}_t = \boldsymbol{J}^{-1} \boldsymbol{e}, \quad \boldsymbol{e} = \boldsymbol{J}_f^{-1} (f(\hat{\boldsymbol{x}}_0(\boldsymbol{x}_t)) - \boldsymbol{y}), \tag{7}$$

assuming the inverses exist. In the case of $f(\boldsymbol{x}) = \boldsymbol{A}\boldsymbol{x}$ for inpainting and super-resolution tasks, where $\boldsymbol{y}$ is lower-dimensional, the inverse of $\boldsymbol{J}_f$, is not defined, but we can use the pseudo-inverse. Similarly, for a non-linear $f$, we can compute $\boldsymbol{e}$ using numerical methods, e.g. backpropagation through $f$ when it is a neural network, which would approximate $\boldsymbol{e}$ using $\boldsymbol{J}_f^T$.

Therefore, $\boldsymbol{e}$ is the same in gradient descent and Newton optimization, but its relationship to $\boldsymbol{e}_t$ differs

$$\text{(a) GD}: \ \boldsymbol{e}_t = \boldsymbol{J}^T \boldsymbol{e}, \quad \text{(b) Newton}: \ \boldsymbol{J} \boldsymbol{e}_t = \boldsymbol{e}. \tag{8}$$

Gradient descent can be computed without directly evaluating the Jacobian by backpropagation on the scalar cost $C$. For the computation of $\boldsymbol{J}^{-1} \boldsymbol{e}$, on the other hand, we have no such method.

In cases where computing the inverse of the Jacobian is prohibitive, inexact Newton methods [7] propose first finding an approximation $\boldsymbol{e}_t^*$ to the solution of Eq (8) (b), performing the Newton step using the approximate solution, and reiterating until convergence. When the residual $\boldsymbol{r}$ is strictly reduced at every step, inexact Newton methods converge to the correct solution

$$\boldsymbol{r} = \boldsymbol{e} - \boldsymbol{J} \boldsymbol{e}_t^*, \quad \frac{\|\boldsymbol{r}\|_2}{\|\boldsymbol{e}\|_2} < \eta, \ \eta \in [0, 1). \tag{9}$$

Since we know that the input and output of the denoiser are related by additive Gaussian noise, we propose the inexact move

$$\boldsymbol{e}_t = \boldsymbol{J} \boldsymbol{e} \tag{10}$$

where instead of computing the Jacobian inverse, we use the Jacobian-error vector product. If the Jacobian tells us how to transform a local change in the input to a change in the output, then we posit that, for denoising diffusion models, we can use the same transformation for the inverse. Substituting $\boldsymbol{e}_t^*$ with the proposed update, adding a learning rate $\lambda$ to control convergence, we get the residual

$$\boldsymbol{r} = \boldsymbol{e} - \lambda \boldsymbol{J}^2 \boldsymbol{e} = (\boldsymbol{I} - \lambda \boldsymbol{J}^2) \boldsymbol{e}. \tag{11}$$

For this residual to be strictly reduced at every iteration of our algorithm, we require

$$\frac{\|\boldsymbol{r}\|_2}{\|\boldsymbol{e}\|_2} = \frac{\|(\boldsymbol{I} - \lambda \boldsymbol{J}^2) \boldsymbol{e}\|_2}{\|\boldsymbol{e}\|_2} \leq \frac{\|\boldsymbol{I} - \lambda \boldsymbol{J}^2\|_2 \|\boldsymbol{e}\|_2}{\|\boldsymbol{e}\|_2} = \|\boldsymbol{I} - \lambda \boldsymbol{J}^2\|_2 < \eta. \tag{12}$$

As long as the spectrum of the denoiser Jacobian matrix is bounded, we can choose a small enough $\lambda$ that strictly reduces the residual. However, the proposed move would converge slowly if the only $\lambda$ allowed were very small. In practice, we find that we can use large learning rates, making the convergence similar or even better than gradient descent. In Appendix A.1, we present an analysis of the spectral properties of the Jacobian matrix of the Stable Diffusion denoiser used in our experiments. Furthermore, in Appendix A.3 we demonstrate the differences between our proposed inexact and the exact Newton method on a small-scale experiment.

The direction of optimization we propose in Eq. (10) has another advantage over the gradient descent update. The direction $\boldsymbol{e}_t$ can be computed numerically to save both on computation and memory compared to using backpropagation on the cost in Eq. (1). To derive the update, consider the function $h(s) = \hat{\boldsymbol{x}}_0(\boldsymbol{x}_t + s\boldsymbol{e})$ where the variable $s$ is scalar. Its derivative at $s = 0$ is

$$\frac{dh}{ds}\bigg|_{s=0} = \boldsymbol{J} \boldsymbol{e}, \tag{13}$$

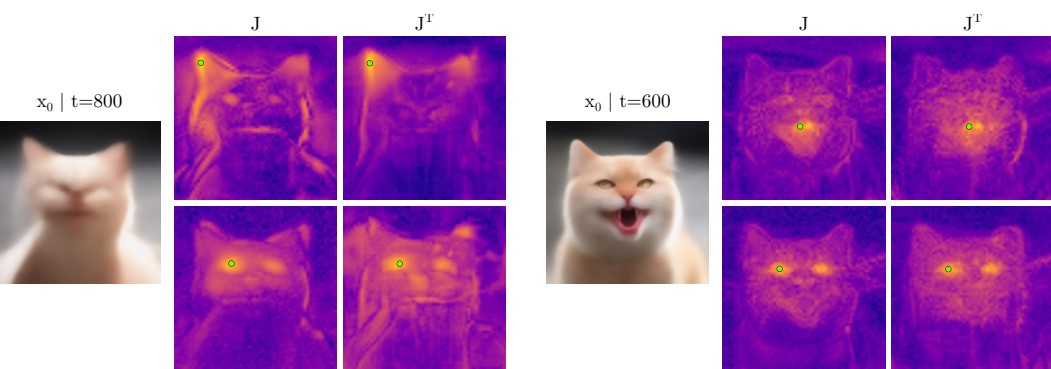

Figure 2: We showcase the difference between the proposed method to compute the update direction ($\boldsymbol{J}$) and gradient descent ($\boldsymbol{J}^T$). The heatmaps indicate where the input $\boldsymbol{x}_t$ would change when perturbing a single pixel in the output, denoted in green. The two directions are considerably different, with ours capturing better longer-range correlations and maintaining shapes. Even though we use finite differences, the direction computed from $\boldsymbol{J}$ is sharper in some regions, like the outlines.

and so the direction $\boldsymbol{e}_t$ can be computed using the numerical derivative of $h(s)$ at $s = 0$

$$\boldsymbol{e}_t = \lambda \boldsymbol{J} \boldsymbol{e} = \lambda \left. \frac{dh}{ds} \right|_{s=0} \approx \lambda \frac{h(\delta) - h(0)}{\delta} = \lambda \frac{\hat{\boldsymbol{x}}_0(\boldsymbol{x}_t + \delta \boldsymbol{e}) - \hat{\boldsymbol{x}}_0(\boldsymbol{x}_t)}{\delta}, \tag{14}$$

requiring no backpropagation but instead two forward passes through the network: one to compute $\hat{\boldsymbol{x}}_0(\boldsymbol{x}_t)$ and another to compute $\hat{\boldsymbol{x}}_0$ for the $\boldsymbol{x}_t$ perturbed in the direction of the error vector.

It is important to point out that the theoretically optimal $\boldsymbol{x}_0(\boldsymbol{x}_t)$ would have a symmetric Jacobian, as $\hat{\boldsymbol{x}}_0$ approximates the true expectation $E[\boldsymbol{x}_0|\boldsymbol{x}_t] = \frac{1}{\sqrt{\alpha_t}}[\boldsymbol{x}_t + \boldsymbol{v}_t \nabla_{\boldsymbol{x}_t} \log p_t(\boldsymbol{x}_t)]$, and the gradient of this is indeed symmetric

$$\nabla_{\boldsymbol{x}_t} E[\boldsymbol{x}_0|\boldsymbol{x}_t] = \frac{1}{\sqrt{\alpha_t}}[\boldsymbol{I} + \boldsymbol{v}_t \nabla^2 \log p_t(\boldsymbol{x}_t)]. \tag{15}$$

This would render the gradient descent and the proposed update equivalent. Yet, we find that the trained denoiser models do not satisfy this ideal condition, making the two update directions noticeably different.

In Figure 2, we visualize this difference between $\boldsymbol{J}$ and $\boldsymbol{J}^T$ by computing rows of $\boldsymbol{J}$ and $\boldsymbol{J}^T$, and averaging the magnitude of all matrix values that contribute to each pixel. This indicates that there is a difference in which pixels in $\boldsymbol{x}_t$ would change when perturbing a single pixel in $\hat{\boldsymbol{x}}_0(\boldsymbol{x}_t)$ for the two update directions. Further experiments on the difference between the two update directions are included in Appendix A.2. Whether the denoising diffusion models are trained with score matching in mind [34] or using the variational method of Ho et al. [16], they do not directly optimize to match the real score $\nabla_{\boldsymbol{x}_t} \log p_t(\boldsymbol{x}_t)$ everywhere nor are they constrained to produce symmetric Jacobians. We hypothesize that this discrepancy in the two update directions is another reason why our optimization of $\boldsymbol{x}_t$ may be more suitable for some applications, which we demonstrate in the experiments (Section 4).

Lastly, the proposed update of Eq. (10) only requires us to provide the direction of the error $\boldsymbol{e}$, which points locally towards a lower cost for the constraint on $\boldsymbol{x}_0$. For linear constraints such as inpainting, this direction can be obtained in closed form using the inverse of the operator expressed by $\boldsymbol{A}$. For non-linear constraints, $\boldsymbol{e}$ does not have to be computed in closed form from $\boldsymbol{J}_f^{-1}$, and can be approximated. In our experiments, we use backpropagation through the differentiable non-linear VAE decoder and constraint, which is cheaper than backpropagating through the denoiser.

An alternative would be to utilize another inexact Newton approximation to avoid backpropagating through the constraint altogether. We discuss this in Appendix A.4, where for linear constraints applied to the the VAE output space (i.e. pixels), we propose an inexact Newton method that utilizes the Jacobian of the VAE encoder to approximate the inverse of the VAE decoder Jacobian. Similar

**Algorithm 1** The proposed algorithm for sampling under linear and non-linear constraints.

---

1: **Input:** Pre-trained diffusion model $\hat{\boldsymbol{x}}_0(\boldsymbol{x}_t)$, linear constraint $C(\boldsymbol{x}_0, \boldsymbol{y}) = \|\boldsymbol{A}\boldsymbol{x}_0(\boldsymbol{x}_t) - \boldsymbol{y}\|_2^2$ or non-linear constraint $C(\boldsymbol{x}_0, \boldsymbol{y}) = \|f(\boldsymbol{x}_0(\boldsymbol{x}_t)) - \boldsymbol{y}\|_2^2$, condition $\boldsymbol{y}$, step size $\delta$, iterations $K$, learning rate $\lambda$, diffusion step size $s$ and schedule parameters $\alpha_i, \beta_i$
2: $\boldsymbol{x}_T \sim N(\boldsymbol{0}, \boldsymbol{I})$
3: **for** $t = T, T - s, T - 2s, \ldots, s$ **do**
4:     **for** $i = 1, 2, \ldots, K$ **do**
5:         **if** Linear **then**
6:             $\boldsymbol{e} = \boldsymbol{A}^T(\boldsymbol{A}\hat{\boldsymbol{x}}_0(\boldsymbol{x}_t) - \boldsymbol{y})$                                     {Using pseudoinverse $\boldsymbol{A}^T$}
7:         **else if** Non-linear **then**
8:             $\boldsymbol{e} = \boldsymbol{J}_f^T(f(\hat{\boldsymbol{x}}_0(\boldsymbol{x}_t)) - \boldsymbol{y}) = \nabla_{\hat{\boldsymbol{x}}_0} C(\hat{\boldsymbol{x}}_0, \boldsymbol{y})$         {Using backpropagation through $f$}
9:         **end if**
10:         $\boldsymbol{e}_t = [\hat{\boldsymbol{x}}_0(\boldsymbol{x}_t + \delta\boldsymbol{e}) - \hat{\boldsymbol{x}}_0(\boldsymbol{x}_t)]/\delta$
11:         $\boldsymbol{x}_t = \boldsymbol{x}_t - \lambda\boldsymbol{e}_t$
12:     **end for**
13:     $\boldsymbol{z}_t \sim N(\boldsymbol{0}, \boldsymbol{I})$
14:     $\boldsymbol{\epsilon}_t = \frac{1}{\sqrt{1-\alpha_{t-s}}}\boldsymbol{x}_t - \frac{\sqrt{\alpha_{t-s}}}{\sqrt{1-\alpha_{t-s}}}\hat{\boldsymbol{x}}_0(\boldsymbol{x}_t)$
15:     $\boldsymbol{x}_{t-s} = \sqrt{\alpha_{t-s}}\hat{\boldsymbol{x}}_0(\boldsymbol{x}_t) + \sqrt{1 - \alpha_{t-s} - \beta_{t-s}^2}\boldsymbol{\epsilon}_t + \beta_{t-s}\boldsymbol{z}_t$            {DDIM step}
16: **end for**
17: **Return:** $\boldsymbol{x}_0$

---

approximations have been discussed in the literature before [36] and can be particularly useful when backpropagation is infeasible [41].

The proposed method for sampling with linear and non-linear constraints is described in Algorithm 1, using DDIM as the diffusion sampling algorithm [33]. Using other diffusion sampling algorithms [19] is intuitive by interleaving the diffusion and our constraint gradient updates on $\boldsymbol{x}_t$ (Appendix A.6).

## 4 Experiments

### 4.1 Linear Constraints

We first verify our algorithm by generating images under linear constraints, which has been the main application of many previous algorithms [5, 28, 6]. We follow the evaluation setting of Saharia et al. [29] and test our method on ImageNet [8], using the first 1000 images from the 10k validation set of the ctest10k split. For evaluation, we measure the PSNR, LPIPS [46], and FID [14] between the real and generated images. We use Stable Diffusion 1.4, which is pre-trained on the LAION [31] text-image pair dataset. Experiments were run on an NVIDIA RTX A5000 24GB GPU.

**Free-form inpainting and $8\times$ super-resolution** In free-form inpainting, masks are randomly sampled and mask out 10-20% of the image pixels. For inpainting with our method, we opt to directly operate on the Stable Diffusion latent given that Stable Diffusion VAE mostly compresses information locally. We apply the masking in pixel space, encode the masked image and inpaint with the unmasked VAE latents. We also apply a $3 \times 3$ dilation kernel on the pixel mask before downsampling, masking out some extra pixels along the edge that we find the VAE fails to encode.

For super-resolution, we cannot apply the constraint in the VAE latent space since image downsampling does not correspond to downsampling VAE latents. This makes super-resolution non-linear. It involves the differentiable decoder $\mathcal{D}$, and to compute the error direction $\boldsymbol{e}$, we backpropagate the pixel-level linear constraint $(\boldsymbol{A}\mathcal{D}(\hat{\boldsymbol{x}}_0)) - \boldsymbol{y})^T(\boldsymbol{A}\mathcal{D}(\hat{\boldsymbol{x}}_0) - \boldsymbol{y})$ through the decoder network. We discuss backpropagation for computing the error direction further in the non-linear constraint experiments (4.2) but leave super-resolution in the linear section as done by previous works.

For inpainting, we set the number of optimization steps $K = 5$ over which we linearly decrease the learning rate $\lambda$ from 0.5 to 0.1. For super-resolution, we use $K = 10$ and a constant $\lambda = 0.1$. For both degradations, we also include additive white Gaussian noise with $\sigma_{\boldsymbol{y}} = 0.05$, use 20 DDIM [33] steps and normalize the computed gradient $\boldsymbol{e}_t$ with its $\infty$-norm.

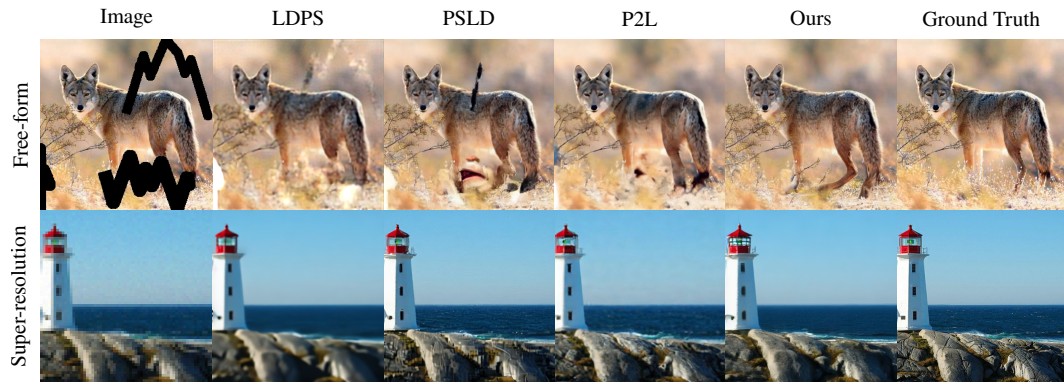

Figure 3: Comparison between our method and existing algorithms on free-form inpainting and $8\times$ super-resolution. We directly use the images and results from [6] since there is no code available to replicate their method.

Table 1: Quantitative evaluation (PSNR, LPIPS, FID) on free-form inpainting.

| | Inpaint (Free-form) | | | |
|---|---|---|---|---|
| **Method** | PSNR ↑ | LPIPS ↓ | FID ↓ | Time |
| P2L[+ captions] [6] | 21.99 | 0.229 | 32.82 | 30m |
| LDPS | 21.54 | 0.332 | 46.72 | 6m |
| PSLD [28] | 20.92 | 0.251 | 40.57 | 8m |
| Ours | 21.73 | 0.258 | 19.39 | 15s |

Table 2: Quantitative evaluation (PSNR, LPIPS, FID) on $8\times$ super-resolution.

| | Super-res ($\times 8$) | | | |
|---|---|---|---|---|
| **Method** | PSNR ↑ | LPIPS ↓ | FID ↓ | Time |
| P2L[+ captions] [6] | 23.38 | 0.386 | 51.81 | 30m |
| LDPS | 23.21 | 0.475 | 61.09 | 6m |
| PSLD [28] | 23.17 | 0.471 | 60.81 | 8m |
| Ours | 24.26 | 0.455 | 60.99 | 1m |
| Ours[+ captions] | 24.95 | 0.405 | 44.74 | 1m |

We present the results on free-form inpainting in Table 1. We find that our method generates better-aligned parts for the missing image regions, reflected in the significant improvement in FID. At the same time, our algorithm maintains consistency with the given image parts, which is reflected in the similar PSNR and LPIPS scores to the baselines.

For $8\times$ super-resolution (Table 2), although the improvements are smaller, we attain similar quality and faithfulness to the generated images at a fraction of the inference time. The best-performing baseline, P2L [6], also utilizes a PaLI VLM [3] to caption the low-resolution images before the diffusion inference. We believe that the main advantage of P2L comes from introducing text conditioning, whereas all other methods rely only on the unconditional model. To verify this assumption, we run our algorithm for $\times 8$ super-resolution with text prompts generated from the downsampled images using Qwen-2.5 [2] as the VLM. The results clearly demonstrate that we can indeed bridge the performance gap to P2L, and even improve upon the PSNR and FID metrics when introducing text captions to the super-resolution process.

Overall, the main advantages are in inference time and GPU memory. We achieve similar or better results to every other method while only requiring a fraction of the compute. When we do not backpropagate through the VAE decoder (inpainting), our method requires only 15 seconds. In super-resolution, where we run more optimization steps and backpropagate, the time increases to 1 minute. In comparison, P2L requires 30 minutes. Since P2L has no public implementation, we report the results directly from the paper and estimate the inference time as $5\times$ that of LDPS [5], which the authors mention as a reasonable expectation. Regarding memory, for a single image, our forward passes only require $\sim 9$GB of memory, compared to backpropagation, which consumes $\sim 17$GB.

**Box Inpainting** In the previous experiments, we observed that our method performed better in inpainting, which required the model to infer more missing information in the given image than in super-resolution. We further push the model by asking it to inpaint a box that covers 25% of the input image. In the comparisons, we also include FreeDoM [44], which, although not previously shown on

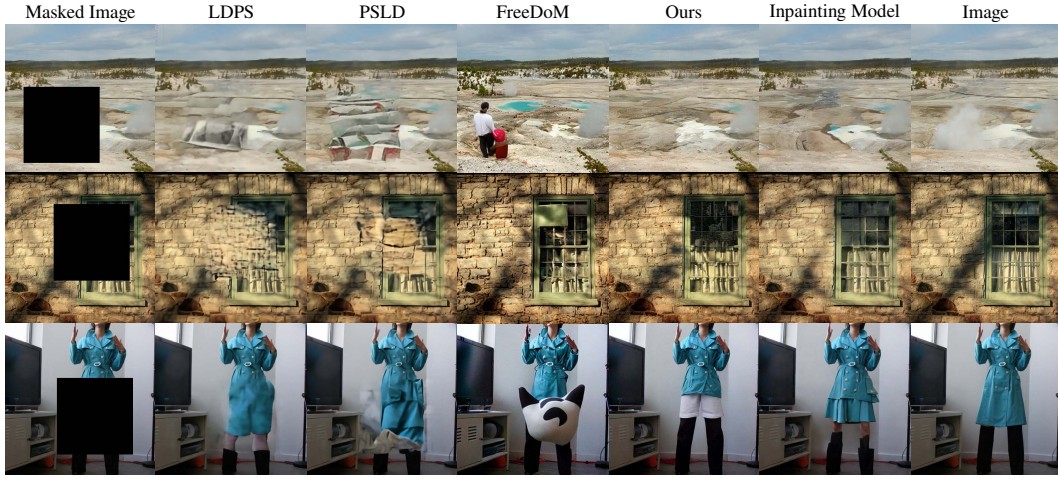

Figure 4: Qualitative evaluation of large area (box) inpainting on ImageNet. Our method achieves results closer to the fine-tuned inpainting model while requiring a fraction of the time to run per image compared to baselines.

inpainting, claims to be a fast training-free inference approach for any condition. We also include the fine-tuned SD-Inpaint model [27], which required 500k additional training steps.

We present the results on box inpainting in Table 3 and Figure 4. Our approach outperforms all existing training-free methods in all metrics and is the closest to the fine-tuned SD-Inpaint model, which we consider an upper limit. Qualitatively, we find that LDPS and PSLD, which perform a lot of denoising steps, generate blurry parts that align with the average appearance over the rest of the image. FreeDoM, the faster baseline, although generating non-blurry parts, frequently fails to maintain consistency with the rest of the image. Our method achieves both high quality and consistency in a shorter time as all previous methods backpropagate through the denoiser weights.

**Number of steps and learning rate**    We ablate the number of optimization steps $K$ and learning rate $\lambda$ hyperparameters of our algorithm on the box inpainting task. The quantitative results are reported in Table 3, where we find that the original choice of $K = 5$ steps and $\lambda = 0.5$ achieve the best reconstruction and image quality metrics. In Appendix Figure 14, we present qualitative results where we observe that running fewer optimization steps gives blurrier results, which is expected as the known regions of the image also seem to not have converged to the given values. Using a higher learning rate leads to the model sometimes 'overshooting' by inpainting the missing regions with realistic-looking parts that do not necessarily fit the rest of the image.

**Finite difference approximation**    In Appendix A.5 we ablate the finite difference step size $\delta$, and compare to the exact Jacobian-vector product computation. We find that our method is robust to the choice of $\delta$ and yields similar results to using the exact computation, while requiring less time.

### 4.2    Non-linear Constraints

Our algorithm can be applied to any non-linear differentiable constraint. The Newton derivation for non-linear constraints in Eq. (7) involves inverting two Jacobian matrices, the denoiser Jacobian $\boldsymbol{J}$ and the Jacobian of the constraint function $f$, $\boldsymbol{J}_f$. Our algorithm offers a way to approximate $\boldsymbol{J}^{-1}$. For $\boldsymbol{J}_f^{-1}$, we use the gradient descent direction $\boldsymbol{J}_f^T$, which is computed using backpropagation through the network $f$. Computing the gradient descent direction for $f$ *does not require backpropagation through the denoiser, only through $f$*. Thus, for non-linear constraints, we combine our proposed Newton direction for the denoiser with gradient descent for the differentiable constraint.

In this section, we showcase style-guided generation with our algorithm. In Appendix B.1, we present results on mask-guided generation, where we show that our method outperforms the strongest baseline, MPGD [13]. We discuss non-differentiable constraints in Appendix B.3.

Table 3: Quantitative evaluation on large area (box) inpainting. Our method outperforms all previous baselines and is the one closest to fine-tuning the diffusion model for inpainting (SD-Inpaint). When using fewer steps $K$, our algorithm does not sufficiently converge. When using a very high learning rate $\lambda$, it overshoots, adding non-realistic parts.

| | **Inpaint (Box)** | | | |
|---|---|---|---|---|
| **Method** | PSNR ↑ | LPIPS ↓ | FID ↓ | Time |
| LDPS | 17.52 | 0.42 | 76.32 | 6m |
| PSLD [28] | 17.30 | 0.38 | 74.02 | 8m |
| FreeDoM [44] | 16.18 | 0.42 | 55.68 | 1m |
| Ours ($K = 5, \lambda = 0.5$) | 18.30 | 0.30 | 42.01 | 15s |
| Ours ($K = 2, \lambda = 0.5$) | 18.01 | 0.39 | 68.75 | 7s |
| Ours ($K = 5, \lambda = 1.0$) | 17.48 | 0.32 | 47.20 | 15s |
| SD-Inpaint | 19.05 | 0.28 | 32.93 | 4s |

Table 4: Quantitative evaluation of style generation. The style score is what the gradient steps are directly optimizing for when using CLIP. Our approach achieves better style scores than the baselines without compromising faithfulness to the prompt, even when using a different model to guide style (OpenCLIP).

| **Method** | Style Score ↓ | CLIP ↑ | Time |
|---|---|---|---|
| DDIM | 761.0 | 31.61 | 9s |
| FreeDoM [44] | 498.08 | 30.14 | 80s |
| MPGD [13] | 441.00 | 26.61 | 50s |
| Ours ($w = 2$) | 368.37 | 23.95 | 45s |
| Ours ($w = 5$) | 310.96 | 24.57 | 45s |
| Ours$^{\text{OpenCLIP}}$ ($w = 5$) | 434.45 | 25.94 | 45s |

| Style | Ours | Ours (OpenCLIP) | FreeDoM | MPGD |
|---|---|---|---|---|

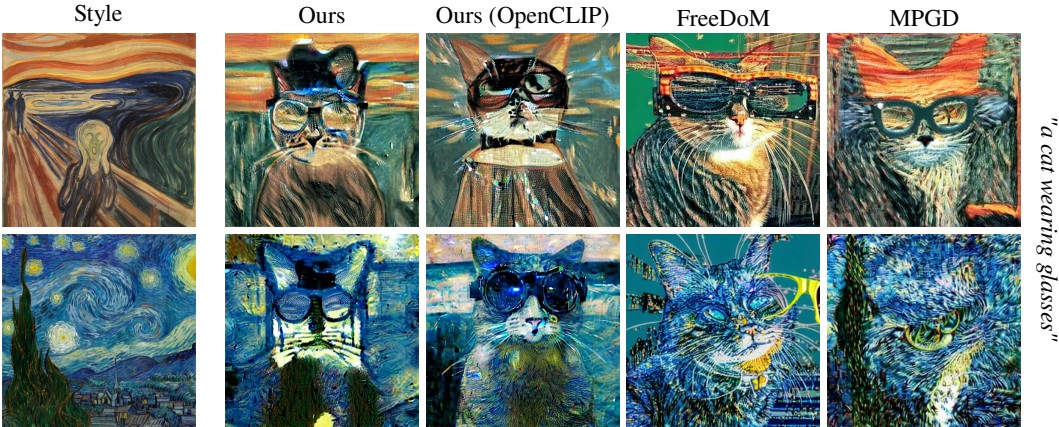

Figure 5: We guide the style of Stable Diffusion images with a CLIP (or OpenCLIP) model, using classifier-free guidance $w = 5$. The images generated by our algorithm are closer to the reference style while maintaining faithfulness to the text prompt.

**Style-guided generation**   The goal is to generate an image that simultaneously follows the style of a reference image $x_{ref}$ and a given text prompt. Following previous works [44, 13], which also perform style-guided generation, we match the statistics of CLIP [25] features between the reference and the generated images while denoising with a text prompt. We use the 2nd CLIP layer features to define the cost $C$ as the Frobenius norm of the the Gram matrix difference [11] $C = \|\text{Gram}(\text{CLIP}_2(x_{ref})) - \text{Gram}(\text{CLIP}_2(\hat{x}_0))\|_F^2$.

Adhering to the evaluation of Yu et al [44], we use 1000 random pairs of reference style images from WikiArt [37] and prompts from PartiPromtps [43]. We measure the CLIP similarity between the generated images and the text prompts to evaluate the faithfulness to the text condition, and the final difference between the Gram matrices to evaluate the faithfulness to the style reference (style score). We use the CLIP ViT-B/16 model for guiding the style of the image and evaluating. We also repeat the experiment using the OpenCLIP ViT-B/32 model [4] for guidance. We perform $K = 5$ gradient updates for every denoising step, using a linearly decreasing learning rate $\lambda$ from 0.5 to 0.1 and classifier-free guidance [15] $w = 2$ and $w = 5$ for the denoiser.

The results are presented in Table 4 and Figure 5. Our method is best at minimizing the constraint, which is the style score evaluation metric. Although in general, the lower the style score the more difficult it is to maintain high CLIP similarity with the prompt, we observe that our method balances

well between the style and text, especially when we increase the guidance weight. When using an OpenCLIP model to guide the style, we achieve a better style score than the baselines that optimized directly for it with CLIP. This also indicates that we minimize the target cost without generating adversarial artifacts that trick the CLIP evaluation.

Excluding the non-guided DDIM, our method is as fast as MPGD [13], which does not backpropagate the style loss through the denoiser network, but only modifies the $\hat{x}_0$ estimation at every denoising step. By only adjusting the $\hat{x}_0$ prediction, MPGD completely fails to propagate the constraint to distant pixels (Appendix B.7), making it unusable in other constrained sampling settings.

**Mask-guided generation**   We guide the Stable Diffusion model with a separately trained face segmentation network. We employ an off-the-shelf model and set the constraint $C$ to be the KL divergence between the per-pixel segmentation classes predicted for a reference image and the generated image. Using 100 images from the FFHQ [17] validation set, we run both our method and MPGD [13], which is the fastest baseline that works very well with 'dense' constraints, i.e. constraints that are applied to all pixels.

The results in Table 5 show that with a similar compute budget, our method achieves both faithfulness (mIoU between generated and reference images) and image quality (CLIP-FID). We used CLIP-FID because it performs better than Inception-FID on a small set of images [18]. MPGD with a high weight ($\rho$) 'burns in' the segmentation mask, leading to artifacts and non-realistic images, whereas a lower weight does not produce images faithful to the mask.

Table 5: Mask-conditioned generation using 100 FFHQ validation set images as reference, using Stable Diffusion with the prompt *'a headshot photo'*.

| Method | mIoU ↑ | CLIP-FID ↓ |
|---|---|---|
| DDIM | 0.09 | 48.78 |
| MPGD [13] ($\rho = 1$) | 0.47 | 77.11 |
| MPGD ($\rho = 0.5$) | 0.36 | 56.38 |
| Ours | 0.42 | 59.79 |

We provide qualitative results of our segmentation mask-guided generation experiment in Appendix Figure 13. The baseline (MPGD) either over-satisfies the constraint by burning in artifacts ($\rho = 1$) or fails to generate images that adhere to the constraint ($\rho = 0.5$). Our approach generates the most realistic images while also getting the mask prediction to match to the reference image. Using Stable Diffusion, we chose the text prompt *'a headshot photo'* to constrain the generated images. Considering the limitations of generating faces with Stable Diffusion, the results may not be on par with an FFHQ-specific model, but we still find our algorithm able to synthesize more usable images than the baseline, even in this difficult case.

**Limitations**   Using high classifier-free guidance ($w > 10$), which is useful in some training-free tasks such as multi-view inference [42], requires us to significantly reduce the learning rate $\lambda$. The high guidance alters the spectral properties of the denoiser Jacobian, making the model more sensitive to small changes in the input. Another limitation we highlight is with distilled models [35, 21] that perform inference in fewer (1-5) steps. We observed that our algorithm requires more steps to achieve comparable results (Appendix B.4), mitigating the inference speed benefits of distilling the model.

## 5   Conclusion

We presented a new algorithm for inference under arbitrary constraints in pre-trained diffusion models. Our approach exploits the relationship between the noisy image input and clean image output of the denoiser to approximate the Newton optimization steps with cheap forward passes. The images generated under linear and non-linear constraints are comparable to or better than state-of-the-art methods, at a fraction of the inference time. We offer a practical algorithm to sample from large pre-trained generative image models under any condition, with the potential to enable new training-free downstream applications that rely on a strong image prior.

## Acknowledgments

Part of this work was done during an internship at Microsoft Research Redmond. This research was also partially supported by NSF grants IIS-2123920, IIS-2212046.

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

# A  Further analysis of the proposed algorithm

## A.1  Spectra of the denoiser Jacobian

In the main text, we resorted to inexact Newton methods [7] to analyze the convergence of the proposed algorithm. We reiterate that the original Newton (or Gauss-Newton) method aims to solve the system

$$\boldsymbol{J}\boldsymbol{e}_t = \boldsymbol{e}. \tag{16}$$

For a general discussion on how to solve this system of equations we refer the reader to Gavin [12]. In our case of denoising diffusion, where computing the inverse of $\boldsymbol{J}$ is expensive, inexact Newton methods utilize an approximate solution $\boldsymbol{e}_t^*$ to Eq (16), which is guaranteed to converge if the residual is strictly reduced at every step

$$\boldsymbol{r} = \boldsymbol{e} - \boldsymbol{J}\boldsymbol{e}_t^*, \quad \frac{\|\boldsymbol{r}\|_2}{\|\boldsymbol{e}\|_2} < \eta, \ \eta \in [0, 1). \tag{17}$$

When substituting the proposed update $\boldsymbol{e}_t^* = \lambda \boldsymbol{J}\boldsymbol{e}$ (Eq. 10) we get

$$\frac{\|\boldsymbol{r}\|_2}{\|\boldsymbol{e}\|_2} = \frac{\|(\boldsymbol{I} - \lambda \boldsymbol{J}^2)\boldsymbol{e}\|_2}{\|\boldsymbol{e}\|_2} \leq \frac{\|\boldsymbol{I} - \lambda \boldsymbol{J}^2\|_2 \|\boldsymbol{e}\|_2}{\|\boldsymbol{e}\|_2} = \|\boldsymbol{I} - \lambda \boldsymbol{J}^2\|_2 < \eta. \tag{18}$$

Therefore, we need to show that the spectral norm of the matrix $\boldsymbol{I} - \lambda \boldsymbol{J}^2$ is strictly less than 1 for a correct choice of learning rate $\lambda$.

If $\boldsymbol{J}$ was diagonalizable (or generally normal), then we could directly estimate $\|\boldsymbol{I} - \lambda \boldsymbol{J}^2\|_2$ using the largest eigenvalue of $\boldsymbol{J}$. While we cannot directly assume that $\boldsymbol{J}$ is normal (we explicitly mentioned that $\boldsymbol{J}$ is not guaranteed to be symmetric), we know that $\boldsymbol{J}$ should be 'almost' symmetric, as is the optimal denoiser solution shown in Eq. (15). This means that we could express $\boldsymbol{J} = \boldsymbol{S} + \boldsymbol{K}$, where $\boldsymbol{S} = (\boldsymbol{J} + \boldsymbol{J}^T)/2$ is the symmetric and $\boldsymbol{K} = (\boldsymbol{J} - \boldsymbol{J}^T)/2$ the skew-symmetric component. Since both the symmetric and skew-symmetric components are normal we can estimate their spectral norms. In the case where $\|\boldsymbol{K}\|_2 \ll \|\boldsymbol{S}\|_2$, which we expect since $\boldsymbol{J}$ is 'almost' symmetric, we can approximate $\|\boldsymbol{I} - \lambda \boldsymbol{J}^2\|_2$ as

$$\|\boldsymbol{I} - \lambda \boldsymbol{J}^2\|_2 \approx |1 - \lambda \max_i \mu_i^2| \tag{19}$$

where $\mu_i$ are the eigenvalues of $\boldsymbol{S}$.

To get an estimate of the magnitudes of the largest eigenvalues of $\boldsymbol{S}$, which are real, and of the eigenvalues of $\boldsymbol{K}$, which are imaginary, we use the Arnoldi iteration [1]. For the Arnoldi iteration we only require access to the matrix-vector products $\boldsymbol{J}\boldsymbol{v}$ and $\boldsymbol{J}^T\boldsymbol{v}$, which can be computed using the approximation of Eq. (14) and backpropagation respectively.

Running the Arnoldi iteration algorithm on $\boldsymbol{J}$, $(\boldsymbol{J} + \boldsymbol{J}^T)/2$ and $(\boldsymbol{J} - \boldsymbol{J}^T)/2$ we plot the magnitude of the largest eigenvalue of each matrix in Fig. 6 (a). We see that the contribution of the symmetric part of the matrix is the strongest, validating our assumption that the diffusion model's Jacobian $\boldsymbol{J}$ is 'almost' symmetric.

If we approximate the spectral norm of Eq.( 18) using the symmetric component of the Jacobian we see that for a correct choice of $\lambda$ we can ensure that $|1 - \lambda \max_i \mu_i| < 1$. Now we revisit the example of Fig. 1 where we inpaint half of the image. Instead of using the empirical learning rate used in the paper (linearly decreasing and gradient normalized by the $\infty$-norm) we compute the largest eigenvalue of $(\boldsymbol{J} + \boldsymbol{J}^T)/2$, $\mu$, and select different $\lambda$ so that we satisfy or violate the bound provided by Eq. (19).

In Fig. 6 (b) we show that the error is not reduced for learning rate values that consistently violate the proposed bound. As expected, our algorithm consistently reduces the error at every iteration for a small enough learning rate, where the bound is always satisfied. When we use the empirical learning rate, shown in Fig. 6 (c), we see that the algorithm bounces between satisfying and not satisfying the computed bound and ends up at a lower error than the constant learning rate of Fig. 6 (b). We posit that the normalization by the $\infty$-norm helps the learning rate adjust the step size such that it substantially reduces the error while not leading to divergence.

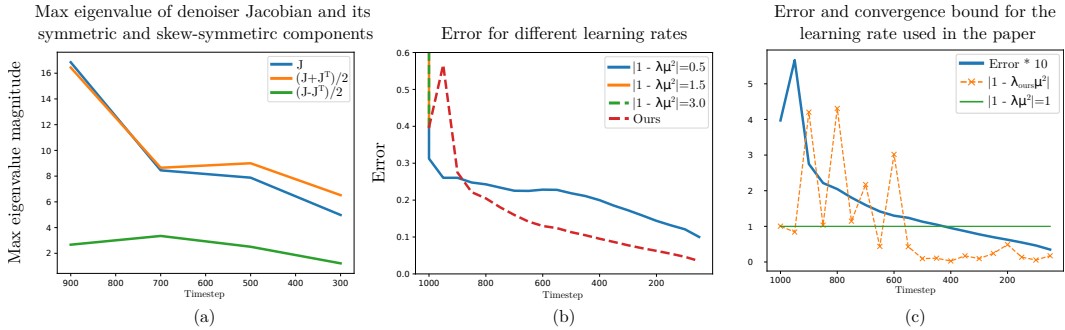

Figure 6: Using the proposed analysis we visualize in (a) the largest eigenvalue of $\boldsymbol{J}$ as well as its symmetric and skew-symmetric components. The largest eigenvalue follows closely the largest eigenvalue of the symmetric part of the matrix. In (b) we demonstrate the convergence of our method for different learning rates. In (c) we show that our adaptive learning rate scheme initially oscillates around the theoretical convergence bound and eventually settles well-below it. These initial oscillations may be important for the high-quality results, as using lower learning rate did not attain similar-quality images.

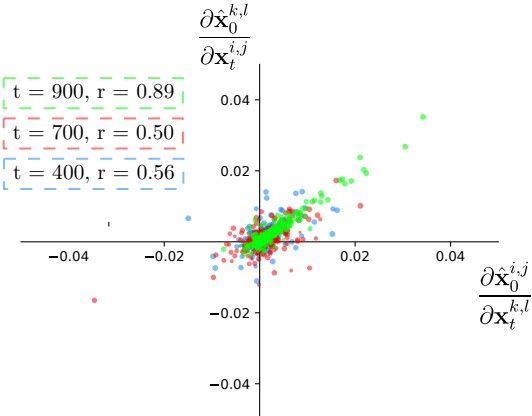

Figure 7: Sample pairs $(i,j),(k,l)$ of the denoiser Jacobian $\nabla_{\boldsymbol{x}_t}\hat{\boldsymbol{x}}_0 = \nabla_{\boldsymbol{x}_t}E[\boldsymbol{x}_0|\boldsymbol{x}_t]$ for different timesteps $t$. We observe that the Jacobian is not symmetric, justifying the difference between the proposed update steps and the previously used gradient descent steps.

## A.2 Symmetry of the Jacobian and difference in updates

**Symmetry** Although, theoretically, the Jacobian of the denoiser should be symmetric (Eq. 15), the trained diffusion model does not exactly match the real score and can yield non-symmetric Jacobians. In Figure 7, we perform a simple experiment to visualize this difference; we select a random image from the ImageNet [8] validation set and employ the Stable Diffusion 1.5 model [23] to denoise at three different noise levels $t = \{900, 700, 400\}$. We encode the image using the VAE encoder, scale it and add appropriate noise to get the intermediate diffusion latent for each timestep. We then give the noisy image to the denoiser network and compute the gradients $\partial\hat{\boldsymbol{x}}_0^{k,l}/\partial\boldsymbol{x}_t^{i,j}$ and $\partial\hat{\boldsymbol{x}}_0^{i,j}/\partial\boldsymbol{x}_t^{k,l}$ for randomly chosen pixels $(i,j),(k,l)$ using backpropagation. When we plot the gradients and compute the correlation coefficient $r$ we observe that the values deviate from $y = x$, which would indicate a symmetric Jacobian.

**Toy experiment** Having shown that the Jacobian is not symmetric, we expect to find differences between the gradient updates of gradient descent (Eq. 2.2) and our proposed update (Eq. 10). To highlight these differences, we set up a toy experiment where we update an identical initial image with the two different directions, under the same condition. The experiment is showcased in Figure 8 and again utilizes the Stable Diffusion 1.5 model. We create a synthetic black-and-white grid image

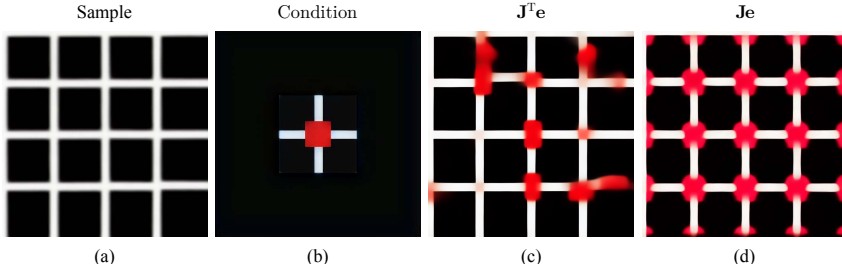

Figure 8: We present a simple experiment to compare the proposed gradient update $\boldsymbol{J}e$ with that of previous methods $\boldsymbol{J}^T e$. The goal is to update a noisy version of the image shown in (a) at $t = 800$, such that the diffusion model's prediction of the final image includes a red square in the middle, shown in (b). The results of gradient descent in (c) and our gradient update in (d) demonstrate that updates performed by previous methods lead to a considerably different result than the update we propose in this paper.

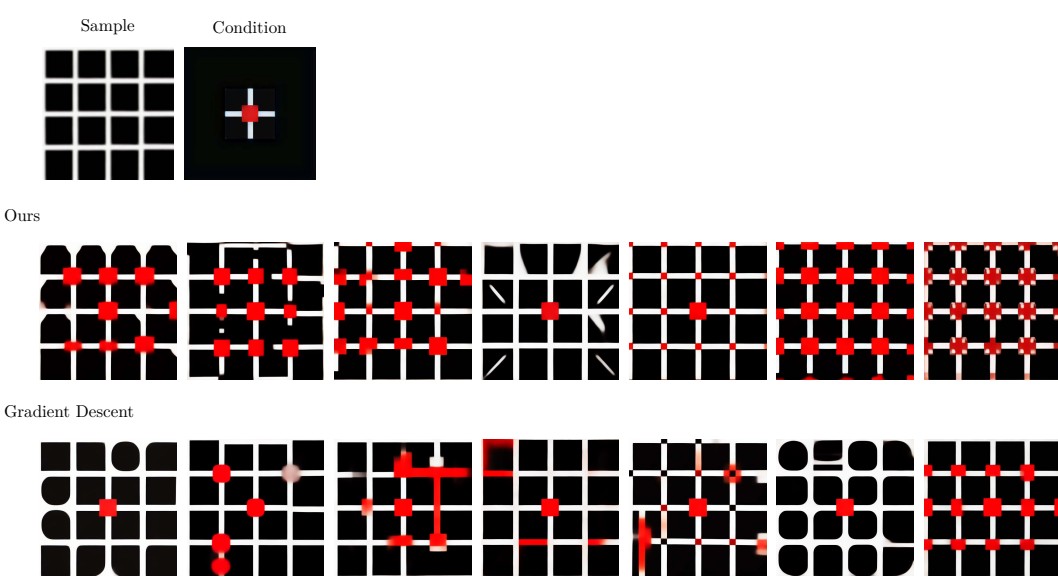

Figure 9: Further runs of the toy experiment of A.2.

(Figure 8 (a)), which we blur, deterministically encode, scale and noise to get a diffusion latent at $t = 800$. We then set up a constraint where we add a red square to the center of the original image (Figure 8 (b)). Instead of decoding the image and applying the constraint in image space, we also encode the constraint image and apply it directly in the Stable Diffusion latent space.

We run 5 gradient updates with the same learning rate of $\lambda = 1$ for each and demonstrate the final predicted $\hat{x}_0$ for the $\boldsymbol{J}^T e$ update of Eq. (2.2) (a) (Figure 8 (c)) and the proposed $\boldsymbol{J}e$ of Eq (10) (Figure 8 (d)). The final result shows how the model intends to change the *entire* image when asked to add a red square in the middle.

The resulting images differ substantially, with the proposed direction producing a more coherent image that tries to copy the newly introduced texture to the correct locations, i.e. the intersections of the lines. Although this is an empirical observation, we hypothesize that the two different directions can have vastly different effects on the image. In Figure 9 we repeat this experiment over multiple random seeds.

**More visualizations** In the main text, we referred to the difference between $\boldsymbol{J}$ and $\boldsymbol{J}^T$ as a motivating factor for our approach and measured that difference by comparing elements $(i, j)$ and $(j, i)$ of the Jacobian matrix (Figure 7). In Figure 10 we extend Fig. 2. We visualize the difference

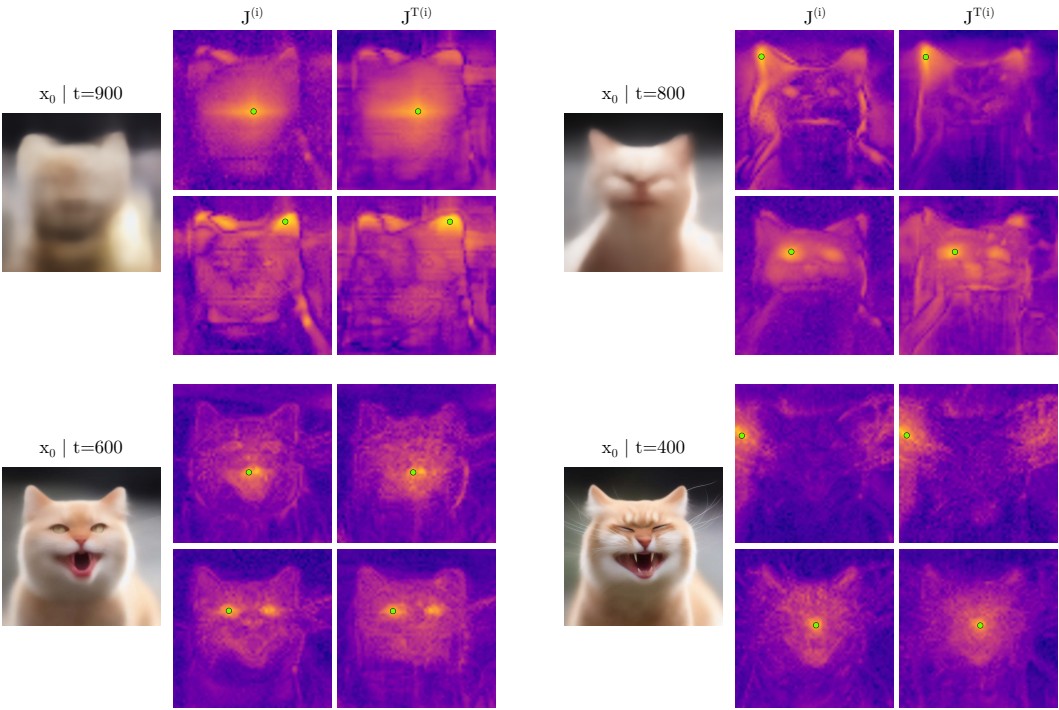

Figure 10: We visualize the difference between $J$ and $J^T$ by computing how the matrix wants to change the entire image would look for a perturbation of a single pixel. By $J^{(i)}$ we denote this exact change, which corresponds to the sum of 4 columns in the matrix (4 latent channels per-pixel). Our proposed update direction better captures longer-range dependencies by better maintaining shapes. Even though we use a numerical approximation, the proposed direction is sharper in regions, like the outline of the cat.

between using the Jacobian and its transpose by numerically computing how the Jacobian and the Jacobian transpose of the entire image would look for a single pixel, i.e. which parts of the image are affected by a change in a single pixel.

In Figure 10 we simplify the notation and denote as $J^{(i)}$ the Jacobian for a pixel $i$, which we compute by summing the squares of the matrix columns that correspond to this pixel's latent values. If our approach were equivalent to backpropagation, i.e. $J = J^T$, then a change in a single pixel would have a similar effect on the rest of the image, up to some noise because of the finite difference approximation.

Contrary to expectations, we find a significant difference between our proposed direction and backpropagation, with our approach having a better effect on retaining shapes and symmetry across the image. We see that in many cases, the model is trying to change correlated parts of the image together, i.e. change both eyes or ears simultaneously, showing us some of the knowledge that the model has acquired regarding the image space in general through its text-to-image training.

### A.3  Inexact and exact Newton

We set up a simple experiment on MNIST to compare the proposed inexact Newton update step with the *exact* Newton, i.e. computing the inverse of the Jacobian. We train a 25M parameter diffusion model, following the architecture of [9], on $32 \times 32$ zero-padded MNIST images. For this model we can use auto-differentiation to compute the $1024 \times 1024$ Jacobian matrix, which takes  10 seconds on our GPUs.

We randomly sample images from the training dataset, add noise corresponding to $t = 700$ and $t = 500$ and denoise, predicting the $\hat{x}_0$. Then, we aim to apply a simple edit to the predicted $\hat{x}_0$ that increases or decreases the value of a single pixel in the image. For that edit, the error $e$ corresponds

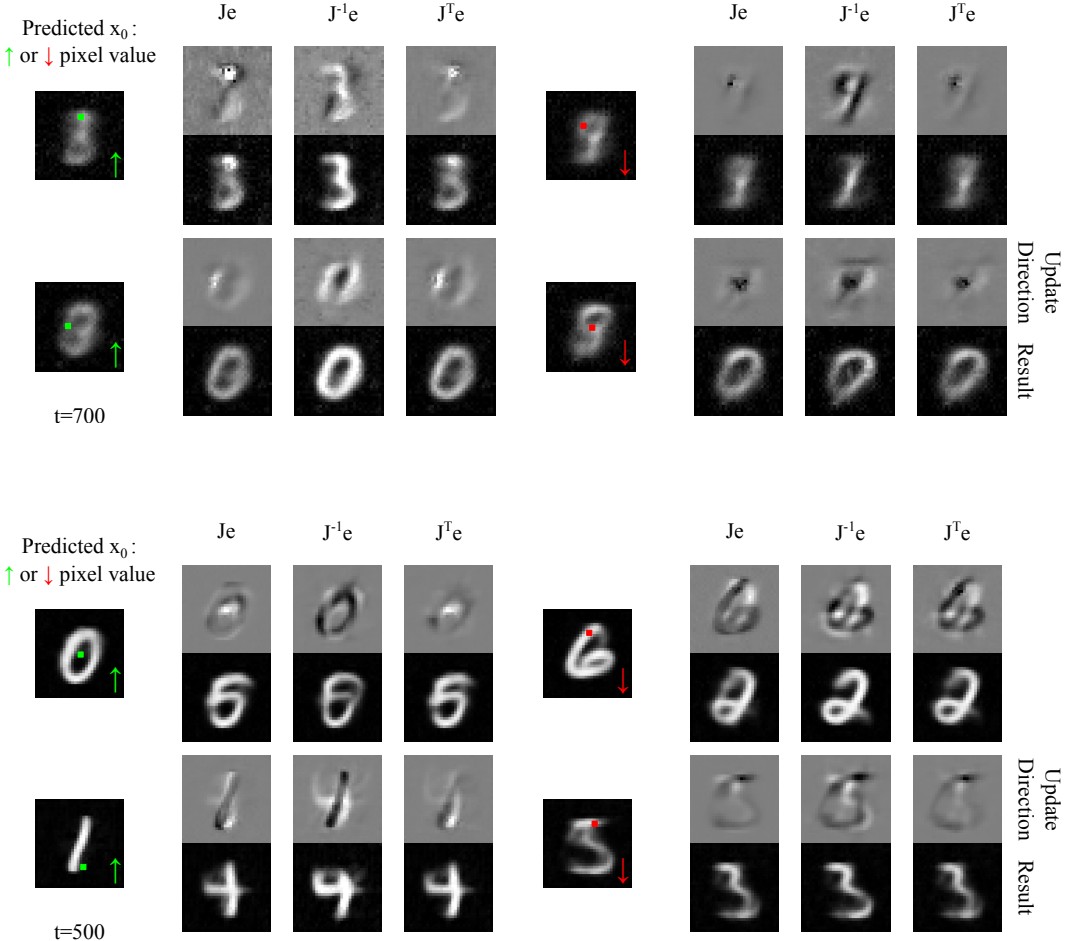

Figure 11: We show the difference between the inexact Newton, exact Newton and gradient descent updates on a simple setting of generating MNIST digits. We simulate a constraint by choosing to increase or decrease the intensity of a single pixel in the image. Qualitatively, the updates made with the inexact and exact Newton methods are similar, with the inexact approach requiring much less compute and being easy to compute in practical settings.

to a +1 or -1 in the location of the edit. For each specific edit, we compute the update we should make to the noisy $x_t$ using our inexact Newton method $Je$, the exact Newton $J^{-1}e$ and gradient descent $J^T e$. For the exact Newton method the Jacobian can be ill-conditioned, requiring us to utilize a pseudo-inverse in computing the update $J^{-1}e$.

In Figure 11 we showcase the results of editing the image with the three update directions. The exact Newton step makes definite steps in updating the image, which converge faster to the desired solution. Our inexact step, qualitatively attempts to make similar edits to the image; e.g. when increasing the pixel intensity inside the digit '0', both updates also delete parts of the side, turning the digit into a 5 (Figure 11, left). Of course, as discussed the main paper, our inexact step requires multiple, smaller steps to achieve the result, but is much cheaper to compute. Finally, the gradient descent direction gives similar results to our proposed update, but as discussed in A.2, showcases qualitative differences that we hypothesize arise from the imperfect training of the diffusion model.

## A.4 Inexact Newton steps in VAE space

In Section 4.2, we described how instead of following the Newton recipe, which requires the inverse of the constraint Jacobian $J_f^{-1}$, we used the gradient descent direction $J_f^T$ when dealing with non-linear

Figure 12: Comaprison between using backpropagation and Newton steps with linear constraints in image space.

constraints. Here, we discuss a special case of non-linear constraints, constraints that are linear in image space. These constraints are still non-linear for latent diffusion models [27] since they involve the VAE decoder, which non-linearly transforms the latent representations that the diffusion model generates to RGB images.

However, when the constraint is linear in image space we find that we can still utilize an inexact Newton approach to avoid backpropagation through the decoder. More specifically, for the case where

$$C(\boldsymbol{x}_t) = (\boldsymbol{A}\mathcal{D}(\hat{\boldsymbol{x}}_0(\boldsymbol{x}_t)) - \boldsymbol{y})^T(\boldsymbol{A}\mathcal{D}(\hat{\boldsymbol{x}}_0(\boldsymbol{x}_t)) - \boldsymbol{y}) \tag{20}$$

we write the first-order Taylor approximation

$$C(\boldsymbol{x}_t - \boldsymbol{e}_t) \approx \boldsymbol{A}\mathcal{D}(\hat{\boldsymbol{x}}_0(\boldsymbol{x}_t)) - \boldsymbol{A}\boldsymbol{J}_{\mathcal{D}}\boldsymbol{J}\boldsymbol{e}_t - \boldsymbol{y})^T(\boldsymbol{A}\mathcal{D}(\hat{\boldsymbol{x}}_0(\boldsymbol{x}_t)) - \boldsymbol{A}\boldsymbol{J}_{\mathcal{D}}\boldsymbol{J}\boldsymbol{e}_t - \boldsymbol{y})^T. \tag{21}$$

When we solve for $\nabla_{\boldsymbol{e}_t}C = 0$ we get

$$\boldsymbol{J}^T\boldsymbol{J}_{\mathcal{D}}^T\boldsymbol{A}^T(\boldsymbol{A}\mathcal{D}(\hat{\boldsymbol{x}}_0(\boldsymbol{x}_t)) - \boldsymbol{y}) = \boldsymbol{J}^T\boldsymbol{J}_{\mathcal{D}}^T\boldsymbol{J}_{\mathcal{D}}\boldsymbol{J}\boldsymbol{e}_t. \tag{22}$$

Similarly to the results in the main paper, assuming that inverses exist, we end up with the following system

$$\boldsymbol{A}^T(\boldsymbol{A}\mathcal{D}(\hat{\boldsymbol{x}}_0(\boldsymbol{x}_t)) - \boldsymbol{y}) = \boldsymbol{J}_{\mathcal{D}}\boldsymbol{J}\boldsymbol{e}_t \tag{23}$$

which we rewrite as

$$\boldsymbol{e}_i = \boldsymbol{J}_{\mathcal{D}}\boldsymbol{J}\boldsymbol{e}_t, \quad \boldsymbol{e}_i = \boldsymbol{A}^T(\boldsymbol{A}\mathcal{D}(\hat{\boldsymbol{x}}_0(\boldsymbol{x}_t)) - \boldsymbol{y}). \tag{24}$$

To apply the proposed Newton approach and get an update direction for $\boldsymbol{x}_t$, we must first solve the system $\boldsymbol{e}_i = \boldsymbol{J}_{\mathcal{D}}\boldsymbol{b}$ for $\boldsymbol{b}$, and then use the approximate solution $\boldsymbol{e}_t = \boldsymbol{J}\boldsymbol{b}$. In the super-resolution experiments we performed in the paper, we opted for the gradient descent approach, which can be expressed as $\boldsymbol{b} = \boldsymbol{J}_{\mathcal{D}}^T\boldsymbol{e}_i$. However, this requires backpropagating through the decoder model which in some cases may be inefficient or altogether unavailable.

As an alternative, we can again resort to inexact Newton and use an 'approximate' inverse to $\boldsymbol{J}_{\mathcal{D}}$, the encoder Jacobian $\boldsymbol{J}_{\mathcal{E}}$. Intuitively, the encoder model performs the inverse operation of the decoder, and therefore we could employ it to 'invert' the decoding operation. Using the encoder allows us to replace backpropagation with forward passes by

$$\boldsymbol{b} = \boldsymbol{J}_{\mathcal{E}}\boldsymbol{e}_i \approx [\mathcal{E}(\mathcal{D}(\hat{\boldsymbol{x}}_0) + \delta\boldsymbol{e}_i) - \mathcal{E}(\mathcal{D}(\hat{\boldsymbol{x}}_0))]/\delta. \tag{25}$$

By combining the Newton step for the VAE space and the Newton step for the denoiser we can run our inference algorithm with no backpropagation operations. In Figure 12 we provide some qualitative

Table 6: Ablation study on the choice of $\delta$ and exact forward-mode auto-differentiation. $K$ is the number of steps, $\lambda$ the learning rate and $\delta$ the finite difference step size. By * we denote the parameters used for the experiment in the main paper.

| Parameters | Inpaint (Box) | | | |
| --- | --- | --- | --- | --- |
| | PSNR ↑ | LPIPS ↓ | FID ↓ | Time |
| $K = 5, \lambda = 0.5, \delta = 0.0005$ | 17.47 | 0.37 | 69.02 | 15s |
| $K = 5, \lambda = 0.5, \delta = 0.005^*$ | 18.30 | 0.30 | 42.01 | 15s |
| $K = 5, \lambda = 0.5, \delta = 0.05$ | 18.32 | 0.31 | 42.44 | 15s |
| $K = 5, \lambda = 0.5, \delta = 0.5$ | 15.80 | 0.34 | 61.40 | 15s |
| $K = 5, \lambda = 0.5$, exact | 18.27 | 0.31 | 44.90 | 76s |

comparisons of super-resolution with a Stable Diffusion model when using backpropagation through the decoder and the inexact Newton step in VAE space. Nevertheless, we opted for backpropagation in our experiments since the time required for multiple forward passes through the encoder and decoder models ended up being the same as backpropagating once through the decoder, and the memory requirements were not prohibitive. In cases where memory is an issue, the 'pure' Newton approach could be an appealing alternative to avoid backpropagating through the decoder model.

Using the the Jacobian of the encoder as an approximation for the inverse Jacobian of the decoder, has been discussed before in the context of autoencoders in Sorrenson et al. [36]. To our knowledge, we are the first to employ this approximation for the diffusion autoencoders.

### A.5 Finite-difference approximation and exact gradient computation

To compute the proposed update $\boldsymbol{J}\boldsymbol{e}$, we use the finite difference approximation $\boldsymbol{J}\boldsymbol{e} \approx (f(\boldsymbol{x} + \delta\boldsymbol{e}) - f(\boldsymbol{x}))/\delta$. In comparison, the gradient descent direction $\boldsymbol{J}^T\boldsymbol{e}$ is *exactly* computed using automatic differentiation, usually implemented as the backward gradient computation, e.g. `e.backward()` in PyTorch.

Some libraries also offer forward-mode auto-differentiation, which directly computes the Jacobian-vector product $\boldsymbol{J}\boldsymbol{e}$. However, forward differentiation is not always implemented or optimized as well as the backward propagation of gradients. In the case of PyTorch, which is what we use for running our experiments, forward mode differentiation is not directly implemented for many of the custom layers of the Stable Diffusion model.

To perform a comparison between our finite difference approximation and the exact forward computation, we resorted to the double backward trick, which computes the forward-mode gradient with two backward calls (`torch.autograd.functional.jvp()` in PyTorch). This is of course expected to be slower and more memory-intensive, but we can use it as a baseline to verify the validity of the finite-difference approximation employed in the paper. For completeness, we also ablated the choice of the step size $\delta$. We repeated the box inpainting task of Table 3 and present the results in Table 6.

In our ablations we find that the finite-difference approximation (i) is robust to the choice of $\delta$, and only fails when using too small (0.0005) and too large (0.5) values, and (ii) performs as well as the exact forward mode auto-differentiation while requiring only a fraction of the time.

### A.6 DDIM and other sampling methods

To apply the proposed method, we modified the DDIM sampling algorithm [33]. We provide a side-by-side comparison to show the difference between the original DDIM and the proposed algorithm. Extending our algorithm to other sampling methods should be intuitive by alternating between gradient updates from our algorithm and the diffusion updates computed with the diffusion sampling algorithm used. In Algorithms 2, 3 we sketch out a pseudo-algorithm for applying the proposed algorithm to any diffusion solver.

Beyond DDIM, we also implemented our method with the PNDM scheduler [19] where we get results indistinguishable from DDIM. Both DDIM and PNDM implementations are provided in the GitHub repository. In Appendix B.4, where we applied our method on rectified flows, we use the

Euler ODE solver. Again, our method is intuitive to apply by interleaving the diffusion updates with our proposed optimization steps.

---

**Algorithm 2** Pseudo-algorithm for sampling using a solver and a pre-trained diffusion model.

1: **Input:** Pre-trained diffusion model $\hat{x}_0(x_t)$, diffusion Solver(), diffusion steps $t_1, t_2, \ldots, t_N$, diffusion schedule $\alpha_i$
2: $x_1 \sim N(\mathbf{0}, I)$
3: **for** $t = t_1, t_2, \ldots, t_{N-1}$ **do**
4:     $z_t \sim N(\mathbf{0}, I)$
5:     $x_{t+1} = \text{Solver}(\hat{x}_0(x_t), z_t, \alpha_t, t+1)$
6: **end for**
7: **Return:** $x_N$

---

**Algorithm 3** Pseudo-algorithm for constrained sampling with a solver, a pre-trained diffusion model and using the proposed algorithm.

1: **Input:** Pre-trained diffusion model $\hat{x}_0(x_t)$, diffusion Solver(), constraint $C(x_0, y) = \|f(\hat{x}_0(x_t)) - y\|_2^2$, condition $y$, step size $\delta$, iterations $K$, learning rate $\lambda$, diffusion steps $t_1, t_2, \ldots, t_N$, diffusion schedule $\alpha_i$
2: $x_1 \sim N(\mathbf{0}, I)$
3: **for** $t = t_1, t_2, \ldots, t_{N-1}$ **do**
4:     **for** $i = 1, 2, \ldots, K$ **do**
5:        $e = J_f^T(f(\hat{x}_0(x_t)) - y)$
6:        $e_t = [\hat{x}_0(x_t + \delta e) - \hat{x}_0(x_t)]/\delta$
7:        $x_t = x_t - \lambda e_t$
8:     **end for**
9:     $z_t \sim N(\mathbf{0}, I)$
10:    $x_{t+1} = \text{Solver}(\hat{x}_0(x_t), z_t, \alpha_t, t+1)$
11: **end for**
12: **Return:** $x_N$

---

# B  Additional results

## B.1  Mask-guided generation

For the mask-guided generation experiment we utilized a pre-trained face segmentation model from huggingface `https://huggingface.co/jonathandinu/face-parsing`. In Figure 13 we provide qualitative results of our segmentation mask-guided generation experiment described in the main text. The quantitative results are provided in Table 5 in the main text.

## B.2  Number of steps and learning rate ablation

We repeat the ImageNet box inpainting experiments with fewer optimization iterations and a higher learning rate. We show qualitative results in Figure 14, where we observe that running fewer optimization steps gives blurrier results, which is expected as the known regions of the image also seem to not have converged to the given values. Using a higher learning rate leads to the model sometimes 'overshooting' by inpainting the missing regions with realistic-looking parts that do not necessarily fit the rest of the image. The quantitative results are presented in Table 3 in the main text.

## B.3  Non-differentiable constraints

There is no restriction on defining the constraint $C$ as long as we can get the direction $e$ towards which we want to push the image $x_0$. In the non-linear constraint case, we resorted to using the gradient descent direction $J_f^T(f(\hat{x}_0(x_t)) - y)$ to avoid computing the inverse Jacobian of $f$. In theory, any direction $e$ that locally minimizes the constraint can be used with the proposed algorithm.

As a toy example, we generate images with pixel values quantized to be either 'on' or 'off' (-1 or 1). The constraint first measures whether a pixel value is positive or negative and then sets the error direction $e$ to $|1 - x|$ or $|-1 - x|$ for every pixel accordingly. This is a non-differentiable constraint for which we can easily compute a local gradient that reduces the cost $C$. Using the prompt 'a photo of a cat', we generate quantized images as shown in Figure 15.

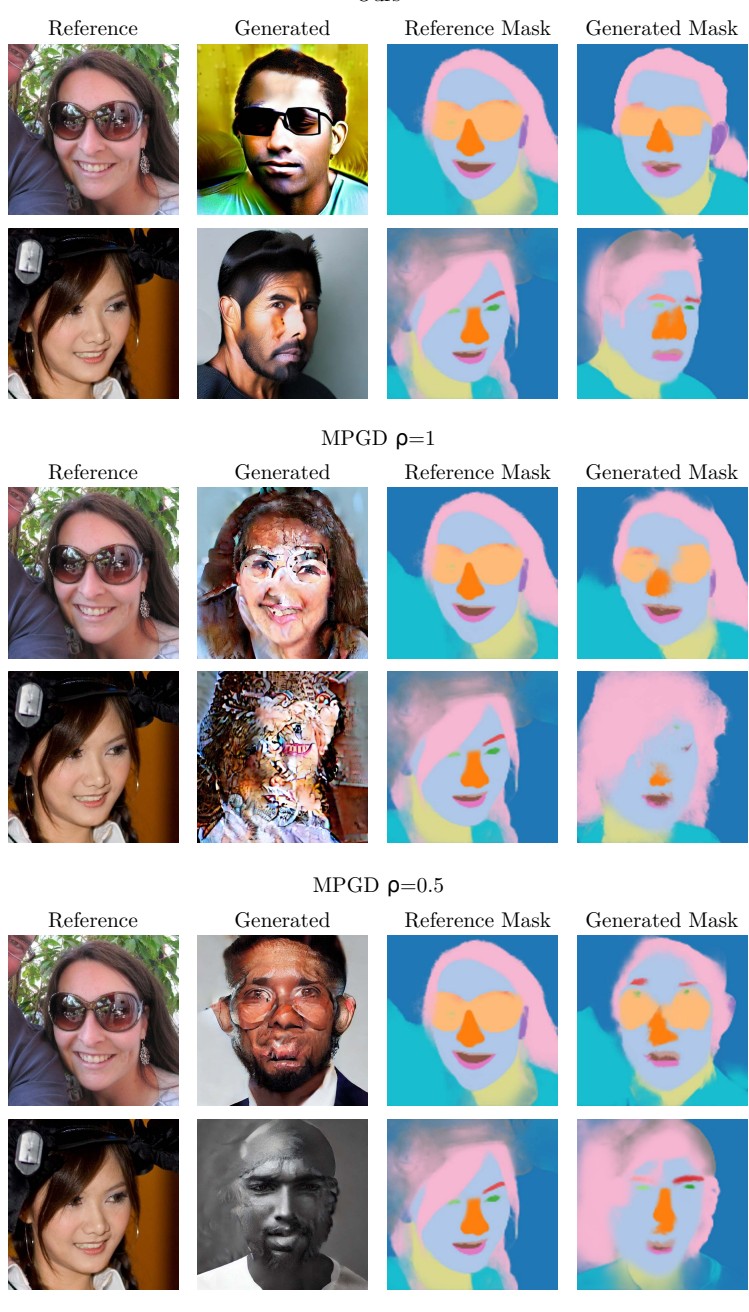

Figure 13: Examples of segmentation-guided generation using our method and MPGD.

## B.4 Time-distilled models

We test whether our method works on distilled models such as rectified flows [20] and consistency models [35]. These techniques reduce the number of inference steps by distilling from a base diffusion model. Our method does not explicitly depend on the number of inference steps used, the type of model or the noise schedule; the only requirement is having a way to estimate the final clean image from the current step, which both rectified flows and consistency models admit. Thus, applying to rectified flows and consistency models is intuitive.

We employ our method to inpaint images using the 2-rectified flow model distilled from Stable Diffusion, InstaFlow [21], using 5 inference steps. We observe that although our algorithm still works

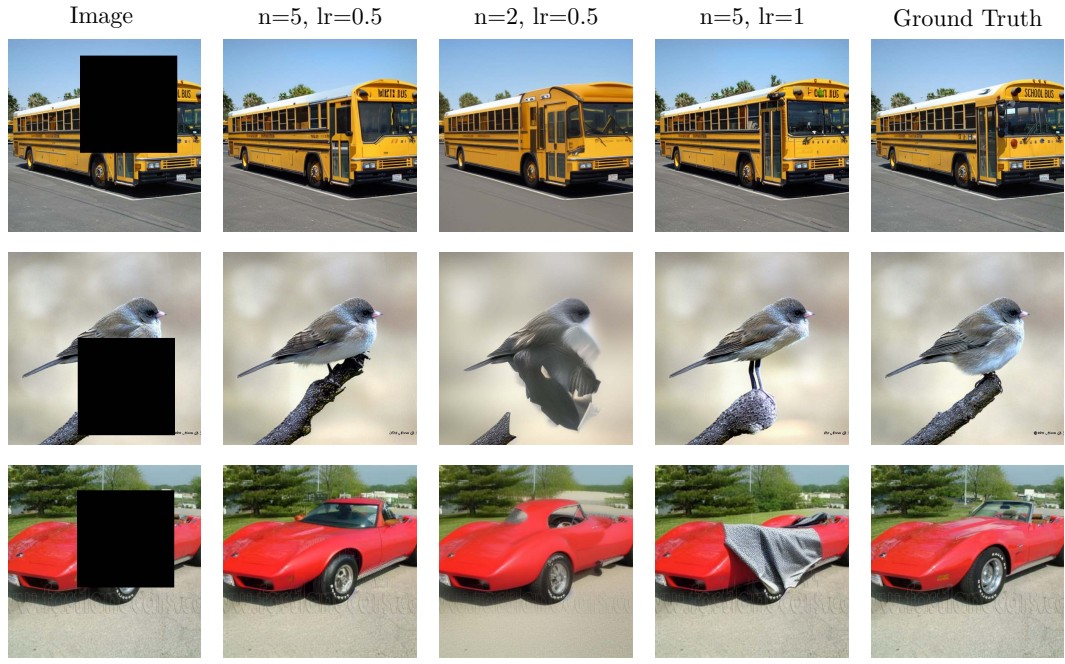

| Image | n=5, lr=0.5 | n=2, lr=0.5 | n=5, lr=1 | Ground Truth |

Figure 14: Examples of box inpainting when using different optimization steps and learning rates.

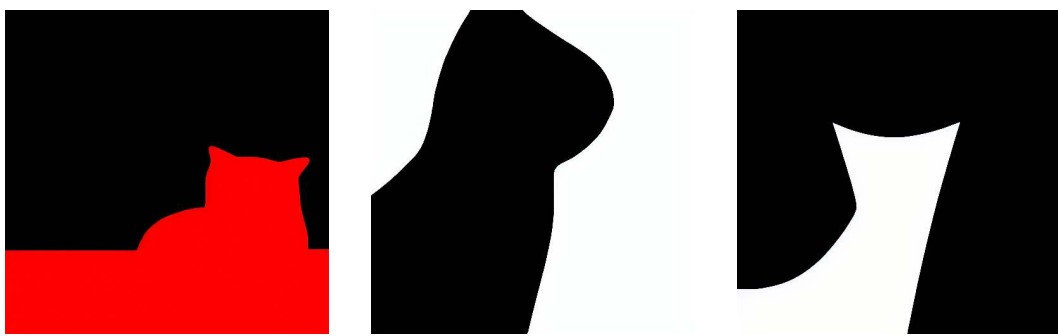

Figure 15: We apply a non-differentiable constraint to generate quantized images. Using the prompt 'a photo of a cat' we generated binary images of cats.

with a rectified flow and just 5 inference steps, we do not consistently get high-quality samples as we did with Stable Diffusion when using the same hypeparameters and it requires more optimization steps. Ultimately, we utilize line search to find the $\lambda$ for every gradient update we perform. This increases inference time further, mitigating the gains from using a time-distilled model.

We hypothesize that using rectified flows (or any distilled model with fewer steps) may be more challenging since the initial noise dictates most of the content in the final image. Therefore, the first optimization steps we perform must get sufficiently close to the correct solution.

When using a diffusion model, we can get away with imperfect optimization steps as there is more room for 'fixing' the image in later timesteps. We show examples of the rectified flow inpainting in Figure 16. We also refer the reviewer to Figures 17,18, where we show the intermediate generation steps for Stable Diffusion.

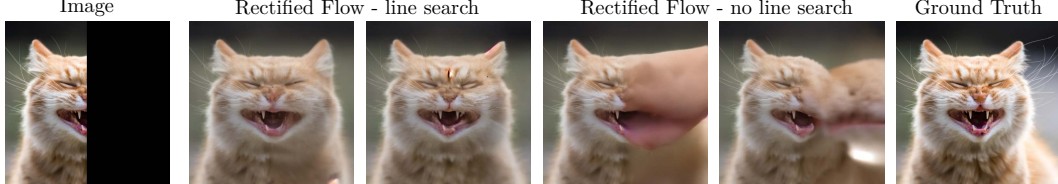

Figure 16: Using a rectified flow model [21] that generates images in just 5 steps requires more careful optimization steps, as there is less room for errors during generation.

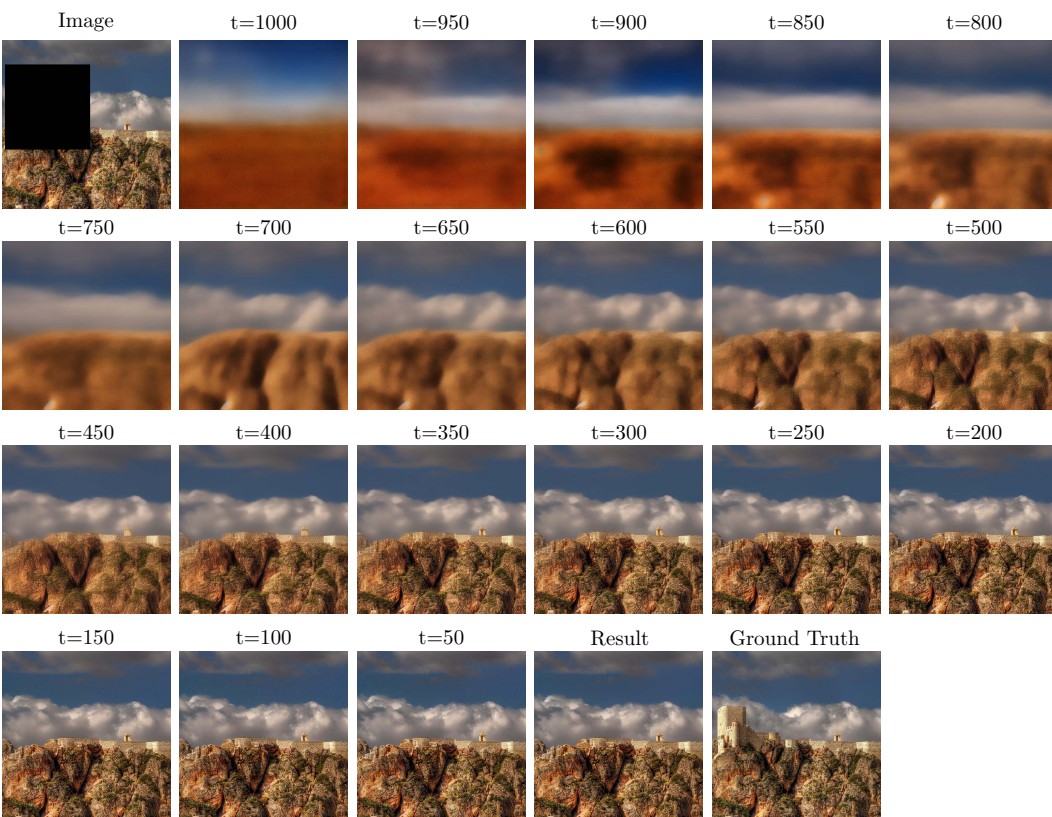

Figure 17: Visualization of the intermediate inference steps for the inpainting task.

## B.5    Inference visualizations

In Figures 17, 18, and 19, we visualize the intermediate steps of the proposed algorithm for the inpainting, super-resolution and style-guided generation tasks respectively. Our method quickly converges to a plausible image and then further refines it to better satisfy the constraint over the diffusion timesteps. For style-guided generation, we see that the structure of the image is defined in the first few initial steps before the specific style provided is applied.

## B.6    Effect of convergence speed on final images

We ask the question of *how does convergence speed affect the quality of the generated images?* Previous works found that applying a high weight on the constraint led to unwanted artifacts in the generated images [5]. We hypothesize that, apart from artifacts in the gradient, 'over-optimizing' for the condition at a given timestep can affect the generation quality. In practice, if we push the initial

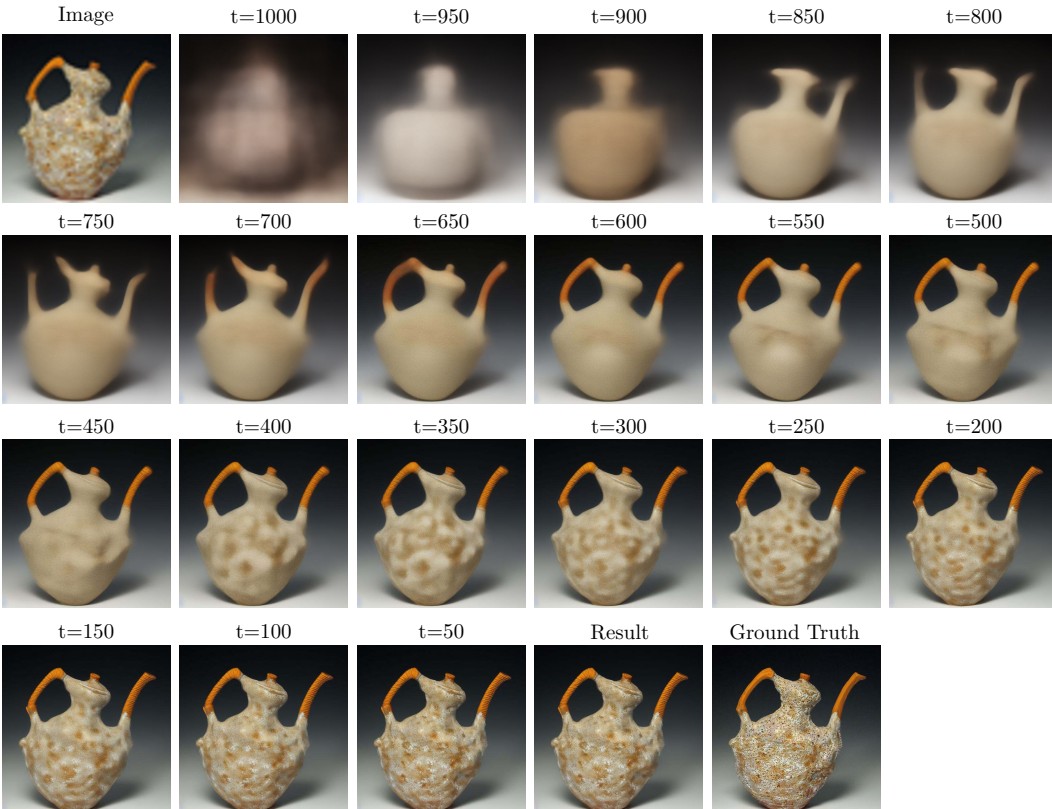

Figure 18: Visualization of the intermediate inference steps for the super-resolution task.

$x_t$ too far from the inputs the denoiser network is expecting, either with a high weight on the gradient update or by performing too many updates, we should be seeing non-realistic images in the output.

We investigate this by running the same inpainting experiment with 5 optimization steps per timestep (Figure 20) and 20 optimization steps (Figure 21). Although we expected to see a difference in the final generated images, we find that both converge to similar quality results. Our proposed optimization steps at a single timestep consider the Jacobian of the denoiser model, which we find acts as 'regularization' and makes it difficult to produce $x_t$ inputs that satisfy the condition 'early'. Even when running the optimization for more steps at a single $x_t$, we see that although the sample converges faster to the desired condition, the denoiser is still able to continue the diffusion process of $x_t$.

### B.7 More Qualitative Results

In Figure 3 we provided qualitative results on free-form inpainting and super-resolution. In inpainting, our model consistently performs as well as the slowest baseline, P2L. For super-resolution, P2L which also infers a prompt seems to generate better-fitting textures for the images. We hypothesize that by also inferring a prompt the high-frequency detail generation is better-guided in the super-resolution task. In contrast, in inpainting, the non-masked pixels contain enough information about the textures that need to be placed around the image.

In Figure 22 we showcase additional results on the box inpainting task. MPGD [13], which does not backpropagate the constraint error through the diffusion model completely fails at inpainting the missing region. We attribute that to the minimal ability to influence pixels that are 'far' from the constraint at lower noise levels without probing the model weights. The *Naive* algorithm replaces the known pixels in the estimated final image at every denoising iteration.

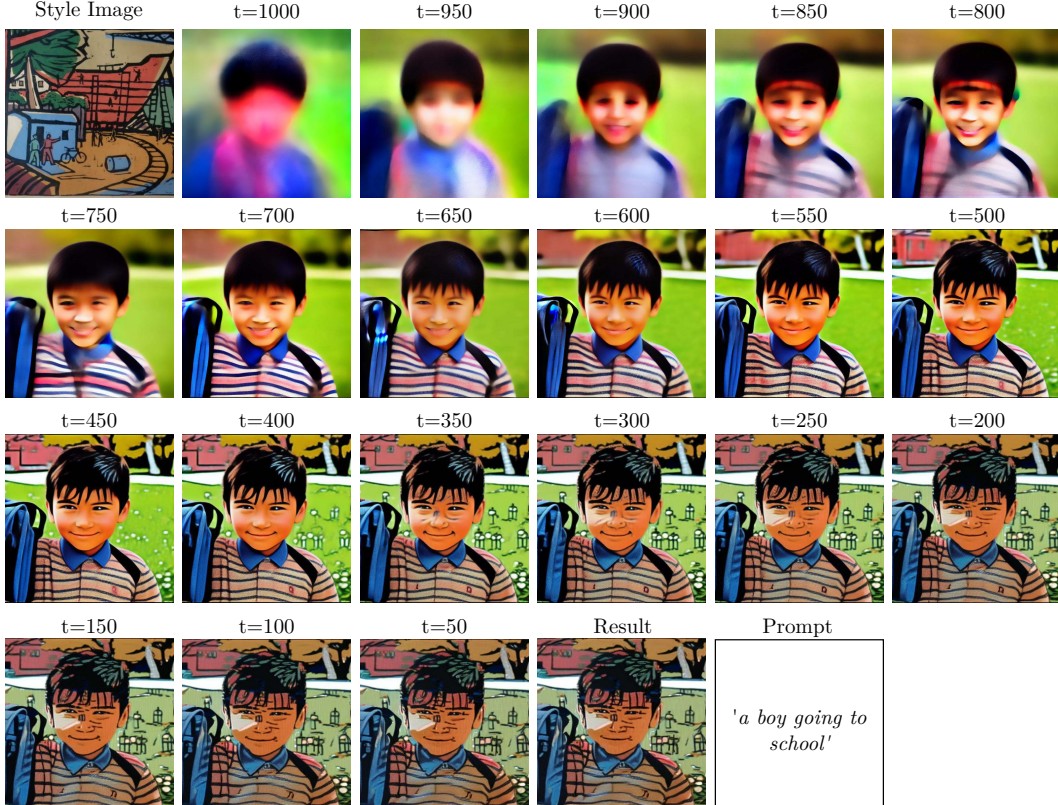

Figure 19: Visualization of the intermediate inference steps for the style-guided generation task.

In Figure 23 we present additional results on style-guided text-to-image generation for a single prompt. Qualitatively, we see that the style of the images generated with our algorithm better matches the style of the reference image, even when using a different model to define style (OpenCLIP). In Figure 24 we show images generated with different styles and text prompts. Here, we show how increasing the classifier-free guidance weight $w$ [15] controls the influence of the text prompt on the final generated image.

## C  Societal Impact

The work presented in this paper aims to advance the field of machine learning, specifically generative modeling. Solving constrained sampling tasks with a generative prior, can greatly benefit from the better utilization of the image prior. One specific domain is compressed sensing in medical imaging, where generative priors like diffusion models have been used to reconstruct low-dose CT scans and accelerated MRIs. We leave to future work the application of the proposed algorithm to these settings.

However, we acknowledge that there are potential societal consequences of our work that can have a negative impact. The one we highlight is the ability to edit images with the intent to deceive and mislead. While it is true that existing models can already be used to alter images, we understand that our work could offer more precise control over the generation and lead to more convincingly fabricated content.

5 steps

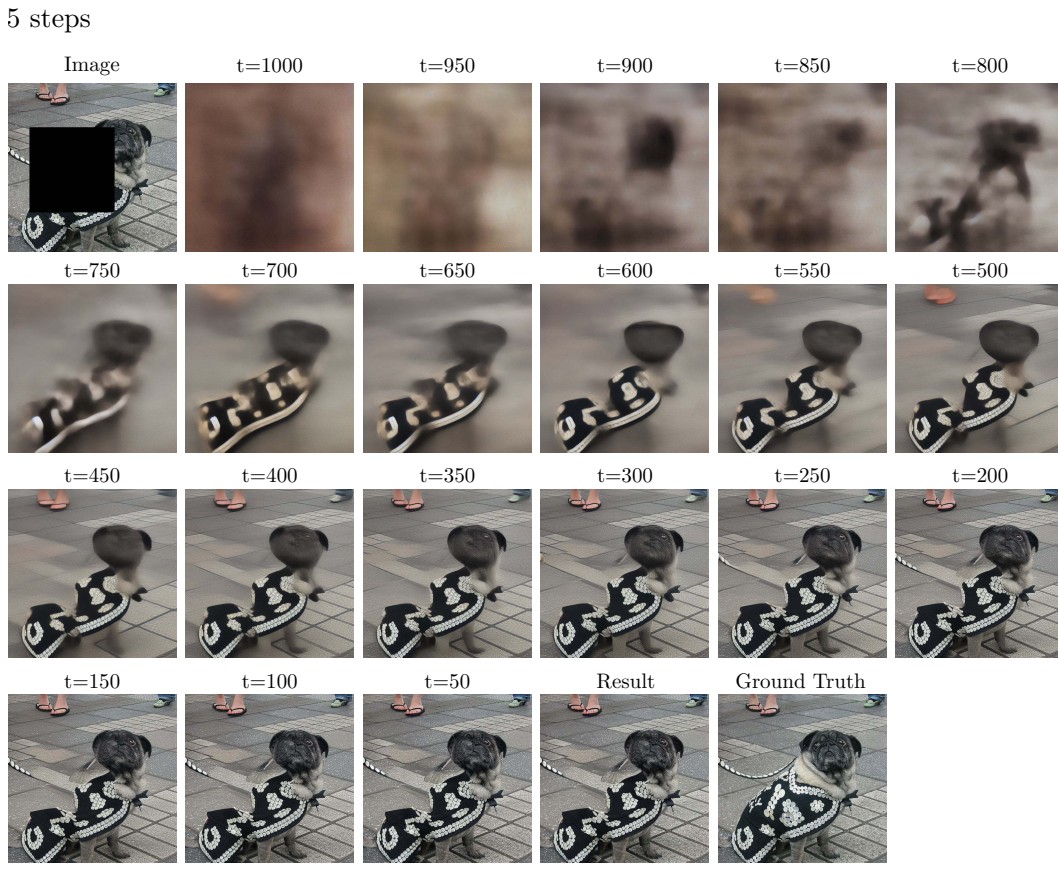

Figure 20: Convergence for the inpainting task when using $K = 5$ optimization steps.

20 steps

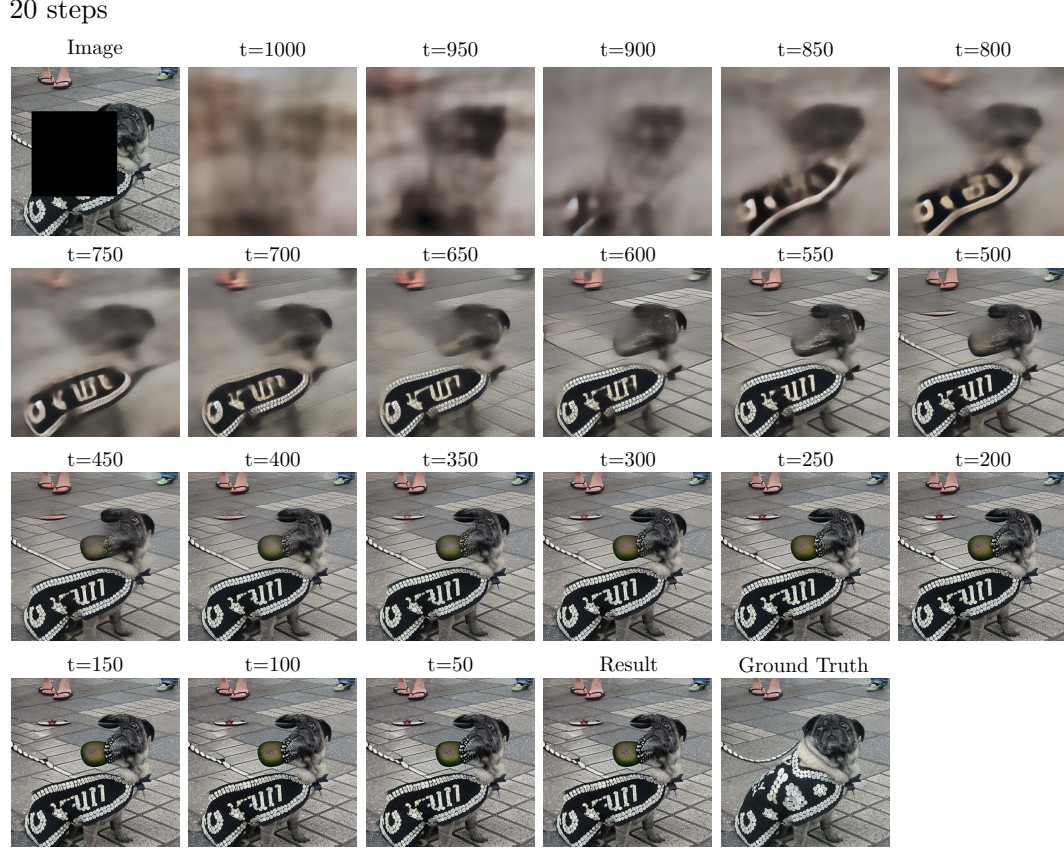

Figure 21: Convergence for the inpainting task when using $K = 20$ optimization steps.

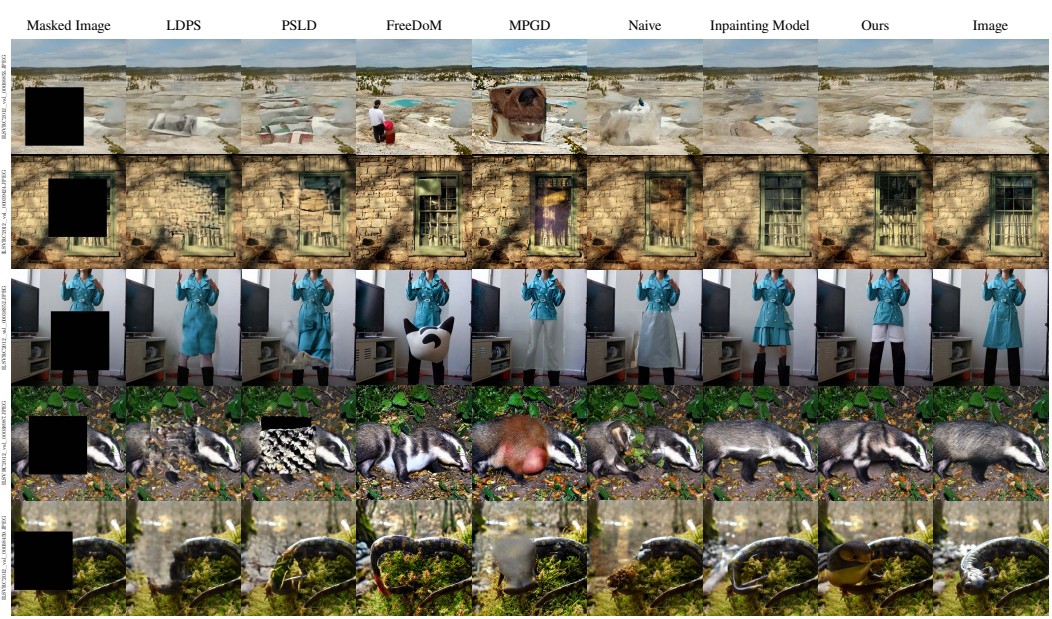

Figure 22: Box inpainting examples for all methods. Naive replaces the pixels with their true values + noise during inference.

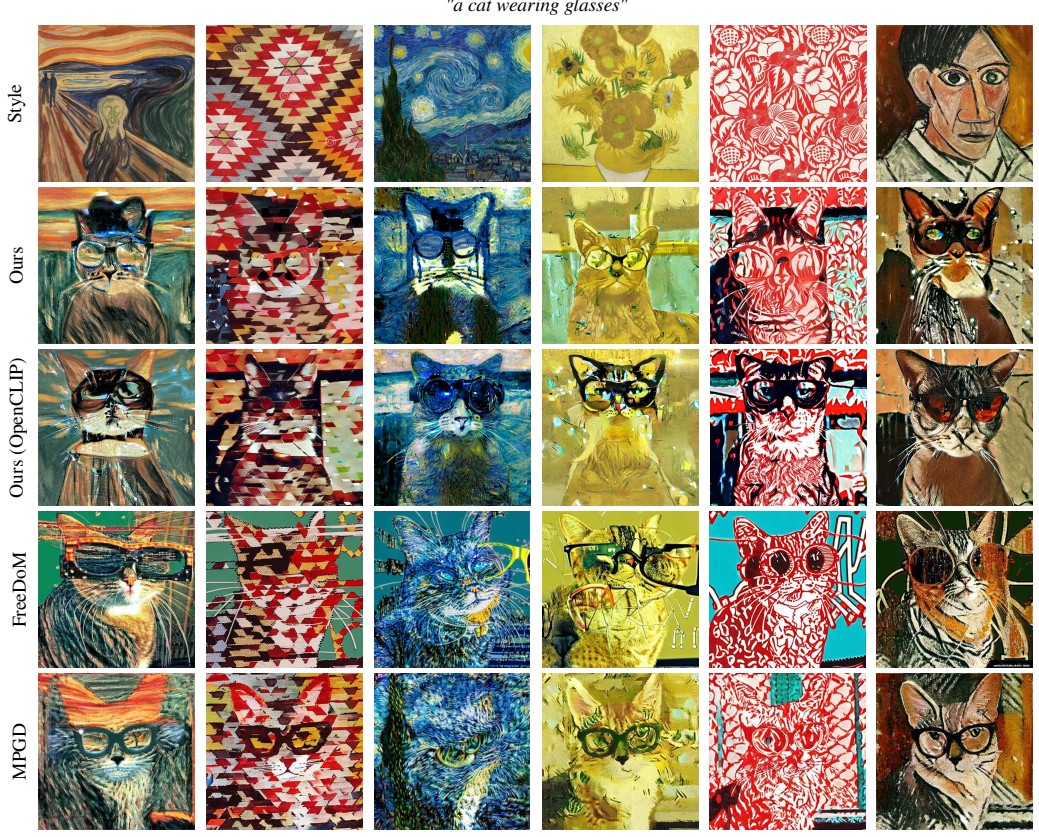

Figure 23: Examples of style-guided text-to-image generation for a single prompt and multiple styles.

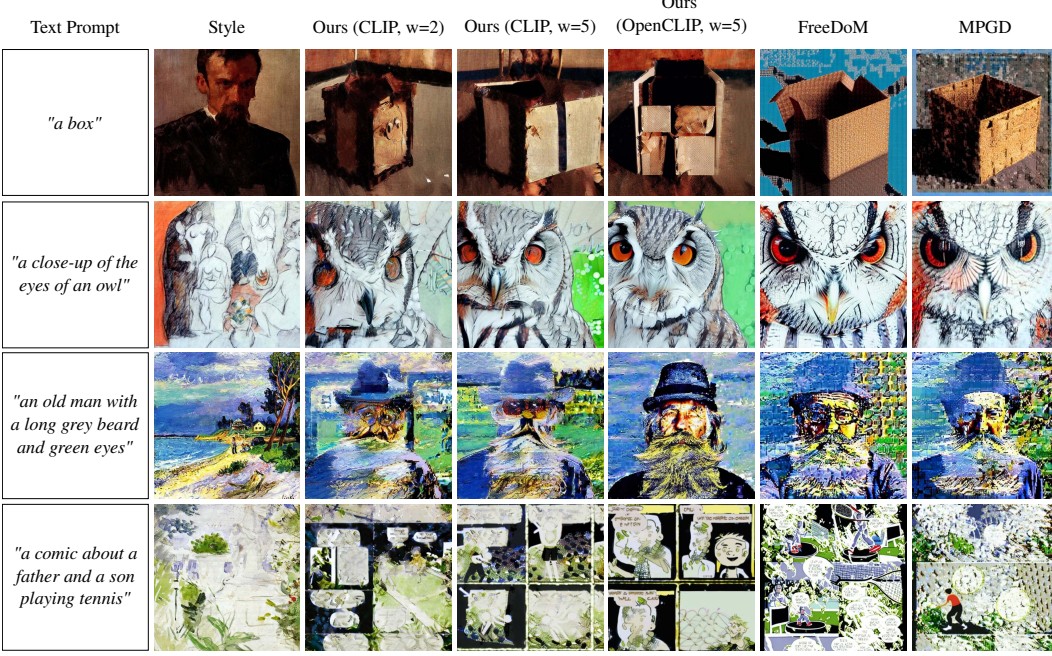

Figure 24: Examples of style-guided text-to-image generation for multiple prompts and styles.

