# OpenReview forum: "Fast constrained sampling in pre-trained diffusion models"
_NeurIPS.cc/2025/Conference — NeurIPS 2025 poster_

### Official Review · Reviewer_sna1 · 2025-06-26

**Clarity:** 2
**Significance:** 2
**Originality:** 4
**Rating:** 4
**Confidence:** 2

**Summary:**

This paper focuses on constrained sampling in diffusion models (e.g., inpainting and super-resolution) and tackles heavy and high-overhead backpropagation in such tasks. Specifically, through rigorous derivation, the authors propose an efficient, high-quality generation paradigm that can operate under arbitrary constraints, which can be regarded as an approximation of Newton's optimization method. Experimental results demonstrate that the proposed approach is effective for inpainting, super-resolution, and style-guided constraint tasks.

**Questions:**

Please refer to the 'Weaknesses' section for details.

My primary unresolved concerns focus on the following aspects:

1. **Writing and Presentations**;
2. **Limited Evaluation Tasks and Necessity**;
3. **Limited Quantitative and Qualitative Performance Gains**.

Overall, while I would consider raising my rating if these concerns are adequately addressed, I think this paper still has significant room for improvement. Although I am not a leading expert in this specific field, based on my knowledge of ​generative models​ and the ​overall quality of this work, I think this paper is ​not yet ready for publication​.

**Ethical Concerns:**

["NO or VERY MINOR ethics concerns only"]

**Final Justification:**

I have carefully read both the other reviewers' comments and the authors' replies. While I'm not an expert in this particular field, I acknowledge that most of my concerns have been addressed. Thus, I have adjusted my rating accordingly.

However, considering the paper's overall quality and contribution, I would suggest the authors carefully address the remaining issues raised by other reviewers, with particular attention to: (i) comprehensive experimental validation, and (ii) clearer presentation or writing logics.

Overall, I think this work currently meets the basic standards for NeurIPS, though it still falls short of being 'exceptional' or 'groundbreaking'."

I recommend placing more emphasis on the feedback from reviewers who specialize in this particular field.

**Limitations:**

Yes, the authors attempt to address the existing limitations.

**Paper Formatting Concerns:**

None.

**Quality:**

2

**Strengths And Weaknesses:**

**Strengths**:

1. **Rigorous Theoretical Derivation**: The paper provides a thorough theoretical analysis and derivation of the proposed method, establishing a solid mathematical foundation.
2. **Computational Efficiency**: Quantitative results in inpainting and super-resolution tasks demonstrate significant efficiency improvements over prior methods, reducing runtime from 6~8 minutes (previous works) to 15 seconds~1 minute.
3. **Performance**: Both quantitative metrics and qualitative results on inpainting and style-guided generation tasks illustrate that the proposed method achieves consistent performance improvements over baseline methods.

**Weaknesses**:

1. **Writing**: While the paper provides meticulous theoretical derivations, the writing style is overly dense and tedious, making it challenging to read. The inclusion of nearly 20 mathematical formulas and pseudocode requires excessive time and effort from readers to comprehend. I recommend that the authors incorporate some highlighted, intuitive summaries to better guide readers through the methodological details and key insights.
2. **Presentation**: Building on the previous point, the manuscript would benefit from adding a framework figure to intuitively illustrate the proposed method, which would help readers quickly grasp the innovation of the approach while making the content more accessible.
3. **Limited Evaluation Tasks**: The paper claims the proposed method is applicable to arbitrary constraints, but the evaluation is restricted to only three tasks (inpainting, super-resolution, and style-guided generation). This narrow scope fails to adequately support the claimed generalizability.
4. **Performance Trade-offs**: As shown in Table 1, while the proposed method achieves remarkable efficiency improvements, it demonstrates advantages only in inpainting FID and super-resolution PSNR metrics. For other metrics, the method falls behind baselines.
5. **Limited Necessity and Performance Gains**: Table 2 reveals that the proposed method underperforms SD-Inpaint in both generation quality and inference efficiency. Although SD-Inpaint requires extra training, this substantial performance gap raises concerns about whether this training-free approach is truly necessary.
6. **Visualizations**: Several concerning artifacts appear in the visual results: In Figure 3 (second row), the proposed method generates images with moiré-like strange textures between the sky and sea surface, which is not found in baseline methods. Moreover, Figure 5 shows that the proposed method produces less natural results compared to baselines, particularly in the first column (Ours), where the generated images appear unnatural with peculiar contours.

---

> ### Author Rebuttal · Authors · 2025-07-30
>
> We thank the reviewer for their time and constructive feedback. We respond to the weaknesses raised and questions below:
>
> **Limited evaluation tasks**:
>
> Our evaluation is performed with both linear (free-form inpainting, x8 super-resolution, box inpainting) and non-linear (style guidance) constraints. This is consistent with previous papers on training-free diffusion sampling algorithms, some of which are often limited to just linear constraints (e.g. PSLD [25], P2L [4]). Additionally,  in Appendix B.1, we included another non-linear task, *mask-guided generation*, which we omitted from the main text due to space constraints. There, we utilize a separately-trained face segmentation network to guide image generation with a target face mask. Any readers interested in mask-guided generation were pointed to it in line 60. Overall, we demonstrated our algorithm on a variety of tasks, showcasing its widespread applicability and consistent advantages over existing baselines.
>
> **Performance trade-offs**:
>
> We argue that we can perform as good as or better than existing training-free sampling algorithms while only requiring a fraction of the inference time. We would like to emphasize that there is **no notable performance trade-off** when using our algorithm, as we discuss below.
>
> - Regarding Table 1, although our algorithm performs considerably worse than P2L in 8x super-resolution, we highlight two key disadvantages of P2L.
>   1. P2L requires 30 minutes to run on a single image, whereas our method only takes one minute.
>   2. P2L uses a VLM (PaLI) to caption the low-resolution images before the diffusion, which introduces additional steps to the inference.
>
>   As mentioned in lines 207-208, we believe that the main advantage of P2L in super-resolution comes from introducing text conditioning. All previous algorithms relied solely on the unconditional model to perform super-resolution, which may not be optimal when the model is required to synthesize specific high-resolution textures in the images.
>
>   To verify this assumption, we ran the x8 super-resolution with text prompts generated from a Qwen-2.5 VLM [Qwen]. We provided the downsampled images and asked the VLM to generate a text caption for each, which we then used as conditioning when running our algorithm. The results below indicate that we can close the performance gap and even improve upon the evaluation metrics when introducing text captions to the super-resolution process. Therefore, there is no trade-off in super-resolution performance when using our algorithm if we also introduce a VLM to caption the low-res images.
>
>   | Method | PSNR ↑ | LPIPS ↓ | FID ↓ |
>   | - | - | - | - |
>   | P2L (with VLM captions) | 23.38 | 0.386 | 51.81 |
>   | Ours | 24.26 | 0.471 | 60.99 |
>   | Ours + VLM captions | 24.95 | 0.405 | 44.74 |
>
> - In Table 2, which presents a more difficult task by inpainting 25% of an image, our method significantly outperforms all baselines and is only outmatched by the fully fine-tuned SD-Inpaint model, which required 100k additional steps of training. Similarly, in Table 3, our method consistently achieves lower style scores compared to the baselines.
>
> To summarize, our method is substantially faster than all baselines and outperforms them on the more challenging tasks. To showcase this better, we will add the text-guided super-resolution result and the relevant discussion to the paper to complement Table 1.
>
> **Limited necessity of training-free sampling**:
>
> We added the SD-Inpaint result in Table 2 as a theoretical upper limit of training-free approaches. We expected that training-free methods would produce worse results than the fine-tuned model, which required considerable additional compute to train. However, this comparison does not take away from the necessity for developing efficient and accurate training-free sampling algorithms:
>
> 1. Training-free sampling algorithms can help understand the inner workings of these massive generative models. In the box inpainting task, we show that, using our method, Stable Diffusion accurately continues the textures from the given parts of an image in the inpainted regions, retaining the overall image structure (Figure 4). These findings can provide valuable insights into how these models understand and process images.
>
> 2. For the linear tasks considered in this (and previous) training-free sampling papers, data can be easily obtained. However, there are cases where there is not enough data to fine-tune a model for conditional sampling. Recent works have applied the training-free sampling algorithms we compare to in our paper, in settings where no trained conditional model exists:
>    - [Reb1] Training-free image generation with image layouts and motion vectors as conditions.
>    - [Reb2] Training-free motion generation with text prompts, keyframes, and obstacles in 3D space as conditions.
>    - [Reb3] Training free molecule generation with geometric constraints.
>
>   Inpainting and super-resolution provide a good testbed for benchmarking training-free sampling algorithms. We show that our algorithm improves upon the performance and efficiency of existing training-free sampling algorithms, and future works can utilize our method as a drop-in replacement for the training-free conditioning guidance.
>
> **Visualizations**:
>
> The artifact mentioned in Figure 3 is a watermark texture hallucinated by the model. Stable Diffusion was trained on web images, which were often watermarked, and can thus generate such patterns. We would like to point out that the hallucinated watermark is adhering fully to the imposed low-resolution image constraint. The mean squared error between the downsampled watermark image and the constraint is around 0.0008, which is also the mean squared error we get when we compare the constraint to another generated lighthouse image that has no hallucinated watermark. We agree that this image may cause unnecessary confusion to the reader and will replace it with another sample that has no watermark hallucination. We will move this example to Appendix B.7 More Qualitative Results.
>
> Regarding Figure 5, the baselines seem to copy the color palette, but do not exhibit the same brush strokes and textures visible in the reference style image. For instance, the tree texture from Van Gogh's painting (bottom row) has been correctly replicated in the cat's body in our generated sample, but is nowhere visible in the baselines. The baseline methods seem to only synthesize a cat fur texture with colors from the reference style.
>
> This qualitative observation translates to the results of Table 3, where the proposed algorithm achieves a lower style score (lower is better), representing the style of the reference image more accurately. If imposing the constraint too strictly generates undesired images, a user can easily relax the constraint by reducing the learning rate $\lambda$ or optimization steps $K$; it is an advantage that our algorithm provides the option to enforce the constraint as strictly as necessary.
>
>
> **Incorporate highlighted, intuitive summaries to better guide readers through the methodological details and key insights**:
>
> The main intuition behind our method is that in diffusion models, we can approximate the Newton optimization step $J^{-1}e$ with several smaller steps in the direction of $\lambda Je$, which is cheap to compute. This approximation is possible in denoising diffusion because $J$ and $J^{-1}$ exhibit similar structures, as discussed in lines 133-134.
>
> To better highlight this intuition, as also suggested by Reviewer NFw9, we performed a smaller-scale experiment, where it is possible to exactly compute the Jacobian and compare $J$ and $J^{-1}$. For a diffusion model trained on MNIST, we showcased how the two matrices encode similar structures by comparing the similarity of their top 20 eigenvectors across different timesteps. Please refer to our response to Reviewer NFw9 for further details on this experiment.
>
> We will include this toy experiment in the main paper, highlighting how we exploit the similar structures of $J$ and $J^{-1}$ to formulate our method. Moving beyond a toy dataset, e.g. a Stable Diffusion-scale model, is computationally infeasible, as a single Jacobian computation would require ~16000 backward passes through the model. Nevertheless, we trust that this toy example can help readers better understand our approach.
>
> **Adding a framework figure**:
>
> Instead of a framework figure, in which it would be difficult to visualize the difference between the various gradient updates in high-dimensional space, we will add examples of the eigenvectors of $J$ and $J^{-1}$ from the toy experiment above. The eigenvectors of $J^{-1}$ are images that showcase how the model encodes correlations between image pixels, i.e., how the model input should change for a constraint applied to its output. By showing that our proposed $Je$ update captures similar correlations to what the optimal inverse update $J^{-1}e$ would, we provide a visual explanation of the intuition behind our method.
>
>
> Additional references:
>
> [Qwen] Bai, Shuai, et al. "Qwen2. 5-VL technical report." arXiv preprint 2025
>
> [Reb1] Wang, Zixuan, et al. "Training-free Dense-Aligned Diffusion Guidance for Modular Conditional Image Synthesis." CVPR 2025
>
> [Reb2] Karunratanakul, Korrawe, et al. "Guided motion diffusion for controllable human motion synthesis." ICCV 2023
>
> [Reb3] Ayadi, Sirine, et al. "Unified guidance for geometry-conditioned molecular generation." NeurIPS 2024

---

> > ### Comment · Reviewer_sna1 · 2025-08-02
> >
> > Thanks for the authors' response and their dedicated efforts.
> >
> > I have carefully read both the other reviewers' comments and the authors' replies. While I'm not an expert in this particular field, I acknowledge that most of my concerns have been addressed. Thus, I have adjusted my rating accordingly.
> >
> > However, considering the paper's overall quality and contribution, I would suggest the authors carefully address the remaining issues raised by other reviewers, with particular attention to: (i) comprehensive experimental validation, and (ii) clearer presentation or writing logics.

---

### Official Review · Reviewer_NFw9 · 2025-06-30

**Clarity:** 2
**Significance:** 3
**Originality:** 3
**Rating:** 4
**Confidence:** 4

**Summary:**

This paper proposes a novel method for training-free constrained sampling. Specifically, instead of backpropagating through the score function (which evaluates whether the constraints are satisfied) and the denoiser, the paper uses an approximate Newton optimization step to avoid backpropagation (though this comes at a cost of one more feedforward evaluation).

Empirical evaluations demonstrate improved performance in image inpainting. It is also faster than some baselines.

**Questions:**

- How does the performance change w.r.t. $\delta$?

- It seems that the proposed method should have similar runtimes compared to the baselines if given the same number of steps. I wonder how the performance compares against the baseline if given the same computation budget?

**Ethical Concerns:**

["NO or VERY MINOR ethics concerns only"]

**Final Justification:**

The rebuttal resolved most of my concerns. However, I do think the authors should give a better explanation on the assumption of $J \approx J^{-1}$ in the next version.

**Limitations:**

Yes.

**Quality:**

3

**Strengths And Weaknesses:**

Strengths:

- The proposed method is training-free and can be adopted to condition on linear and non-linear constraints. The experiments demonstrate good performance on three tasks (image inpainting, super-resolution, and style conditioning).

- The paper is generally well-written and is easy to follow.

Weaknesses:

- The assumption made in Eq. (11) that $J e$ is similar to $J^{-1} e$ needs further empirical/theoretical justification. Although the paper provides a short, intuitive explanation in line 133, I still have concerns regarding whether or how much the assumption holds. I think even empirical justifications (computing both terms and evaluating their difference) would be valuable.

- I didn't really follow the transition from Eq. (10) to Eqs. (12) and (13). Specifically, I was confused by the introduction of the learning rate $\lambda$ since it does not appear in Eq. (10). Could the authors explain more why this can be the case? One explanation could be that $e_t$ is defined as $J e \lambda$. I also had trouble understanding why this is considered Newton's method, since Eq. (11) may be very different than $J^{-1} e$. Even if the analysis is sound, I believe a better explanation should be given in the paper to minimize confusion.

- While the method claims to avoid back-propagation, it seems that Eq. (16) simply replaces auto-diff with a finite-difference approximation of step size $\delta$. This can also be done with forward-mode auto-diff. It would be interesting to also compare their empirical performance.

---

> ### Author Rebuttal · Authors · 2025-07-30
>
> We thank the reviewer for their thorough comments. We respond to the questions below:
>
> **The assumption made in Eq. (11) that $Je$ is similar to $J^{-1}e$ needs further empirical/theoretical justification**:
>
> We understand that the intuition provided in line 133 may not be satisfying. The main difficulty of empirically demonstrating the similarities between $J$ and $J^{-1}$ lies in computing the Jacobian $J$ itself. Computing the Jacobian locally for one input $x_t$ could require ~16000 autodifferentiation passes, one for every row of the Jacobian. Understandably, for a model at the scale of Stable Diffusion (800M parameters), running an experiment where we compute the Jacobian at multiple input points is infeasible within the timeframe of this rebuttal.
>
> As an alternative, we discuss a smaller-scale experiment with a small diffusion model (25M parameters) trained on the MNIST dataset. We padded the MNIST digit images with zeros to 32x32 pixel images, making the input and output of the model 1024-dimensional. Here, $J \in \mathbb{R}^{1024\times1024}$ can be readily computed with automatic differentiation (~10s on our machine), allowing us to also compute $J^{-1}$ assuming $J$ is not singular.
>
> Measuring the similarity between the two matrices is challenging; we find that they differ significantly in scale, with $J^{-1}$ having much larger values than $J$, which makes measuring some norm of their difference (e.g. Frobenius) unreliable. What we are interested in is whether the two matrices exhibit similar structures, i.e., given an input, whether they transform it in similar ways. To measure this, we can compute the eigendecomposition of the two matrices and compare the top-K eigenvectors. If the transformation defined by $J$ is similar to $J^{-1}$, we expect the most important eigenvectors to be similar, which we can easily measure with cosine similarity.
>
> We perform the following experiment: Given a noisy image $x_t$ at timestep $t$, we compute the exact Jacobian $J$ using automatic differentiation (`torch.autograd.functional.jacobian`),  its inverse $J^{-1}$, and the eigenvectors and eigenvalues of both matrices. We then select the top 20 eigenvectors, excluding their imaginary parts, and ordering them by the magnitude of the corresponding eigenvalue. To account for misalignment between the eigenvalues, we pair each eigenvector of $J$ with its closest eigenvector from $J^{-1}$ and measure the average cosine similarity across all eigenvector pairs.
>
> We run this experiment for different timesteps $t$, sampling 10 random images at each timestep from the MNIST dataset to compute $J$ and $J^{-1}$. As a baseline, we choose random $J$ and $J^{-1}$ matrices computed at the same timestep and repeat the experiment to show that the similarities measured are not spurious.
>
> | Timestep $t$ | 950 | 850 | 750 | 650 | 550 | 450 | 350 | 250 | 150 | 50 |
> | - | - | - | - | - | - | - | - | - | - | - |
> | Similarity for the same $x_t$ | 0.68 | 0.72 | 0.62 | 0.72 | 0.71 | 0.72 | 0.72 | 0.63 | 0.60 | 0.58
> | Similarity for random $x_t$ | 0.11 | 0.11 | 0.16 | 0.29 | 0.29 | 0.19 | 0.16 | 0.15 | 0.14 | 0.12 |
>
> By showing the high similarity between the top 20 eigenvectors of the two matrices, we empirically demonstrate the intuition of line 133; the two transformations $J$ and $J^{-1}$ can be used interchangeably to compute the gradient direction for the constraint. A particularly surprising finding is that in some cases, we found almost the same eigenvector (cosine similarity $\approx 1$) from the two matrix decompositions.
>
> Although we demonstrate this in a smaller-scale experiment, we believe it supports our argument in the paper for using an inexact Newton method that approximates the steps of $J^{-1}e$ with $Je$. We will include the above experiment and discussions in the paper to strengthen the intuitive explanation given.
>
> **Eqs. (10)-(13) and introduction of the learning rate**:
>
> Eq. (10) tells us that we must compute an approximate solution $e^\*_t$ that strictly reduces the residual error $r$ if we want the method to converge. The solution we choose in this paper is $e_t^* = \lambda Je$. We agree that it is better to include the learning rate $\lambda$ in Eq. (11) to avoid confusion.
>
> Regarding the role of the learning rate, the sole reason we introduce it is to ensure the convergence of the proposed update. Although in our experiments we tuned the learning rate empirically, in Appendix A we provide an in-depth discussion of how $\lambda$ is related to the spectra for $J$ and affects the convergence. We will point readers to that in the main paper.
>
> **Why is this considered Newton's method?**:
>
> Newton's method involves solving for the inverse of the Jacobian $J^{-1}$ and performing the update $J^{-1}e$. In cases where computing the inverse is infeasible, we can approximate Newton's method with updates $Me$, where $M$ is chosen such that it is as close (in structure) to $J^{-1}$ as possible, but cheap to compute. These methods are called *Inexact* Newton methods and they have been studied in numerical optimization before (see Inexact Newton Methods [5]).
>
> Our algorithm utilizes an Inexact Newton method that approximates $J^{-1}e$ with $Je$. The reason this approximation works well is specific to denoising diffusion models and is discussed above.
>
>
> **Finite-difference approximation and exact gradient computation**:
>
> To compute the $Je$, we use the finite difference approximation $Je \approx (f(x + \delta e) - f(x))/\delta$. The automatic differentiation libraries implement the backward gradient computation, which gives the gradient descent direction $J^Te$. This is what `e.backward()`  computes in PyTorch.
>
> Some libraries also offer forward-mode auto-differentiation, which computes the Jacobian-vector product $Je$. However, forward differentiation is not always implemented or optimized as well as the backward propagation of gradients. In the case of PyTorch, which is what we use for running Stable Diffusion, forward mode differentiation is not directly implemented for many of the custom Stable Diffusion layers.
>
> To perform a comparison between our finite difference approximation and the exact forward computation, we resorted to the double backward trick, which computes the forward-mode gradient with two backward calls. This is, of course, slower and memory-intensive, but we use it as a baseline to verify the finite-difference approximation employed in the paper. For completeness, we also **ablated the choice of $\delta$**
>
> We repeated the box inpainting task of Table 2 and present the results below:
>
> | Method | PSNR ↑ | LPIPS ↓ | FID ↓ | Time |
> | - | - | - | - | - |
> Ours ($K=5$, $\lambda = 0.5$, $\delta = 0.0005$ ) | 17.47 | 0.37 | 69.02 | 15s |
> Ours ($K=5$, $\lambda = 0.5$, $\delta = 0.005$ ) | 18.30 | 0.30 | 42.01 | 15s |
> Ours ($K=5$, $\lambda = 0.5$, $\delta = 0.05$ ) | 18.32 | 0.31 | 42.44 | 15s |
> Ours ($K=5$, $\lambda = 0.5$, $\delta = 0.5$ ) | 15.80 | 0.34 | 61.40 | 15s |
> Ours ($K=5$, $\lambda = 0.5$, exact forward diff ) | 18.27 | 0.31 | 44.90 | 76s |
>
>  $K$ number of steps, $\lambda$ learning rate, $\delta$ finite difference step size
>
> The finite-difference approximation is (i) robust to the choice of $\delta$, and only fails when using too small (0.0005) and too large (0.5) $\delta$ values, and (ii) performs as well as the exact forward mode auto-differentiation.
>
> **Comparing the performance against the baseline with the same computation budget**:
>
> In Appendix B.6, we briefly discuss what happens when the number of constraint optimization steps per denoising diffusion step is increased from 5 to 20. What we observed is that once the constraint optimization converges to a local optimum, the additional steps do not improve the final generated image and may also, in some cases, introduce unwanted artifacts. This is in line with the observations made by previous works (DPS [3]).
>
> We believe that the empirical values for the number of steps $K$ and learning rate $\lambda$ we chose in the paper are adequate for the optimization to converge sufficiently, and thus increasing the computation would not impact the result significantly. Since the results we obtained were already better than the baselines (see our response to Reviewer uiuT for even better super-res results), and our main argument is that training-free inference can be practical (<1min), we did not experiment much with increasing inference time.

---

> > ### Comment · Reviewer_NFw9 · 2025-08-04
> > **thank you**
> >
> > I thank the authors for their detailed response, which clarifies most of my concerns. I hope the authors can incorporate the changes in the next version of the paper. I will adjust my rating accordingly.
> >
> > I have one remaining question. If I understand correctly, the set of eigenvectors for $J$ and its inverse is the same. It seems to imply interesting properties of the eigenvalues, given the observation that "most important eigenvectors to be similar". Could the authors please comment on this?

---

> > > ### Author Response · Authors · 2025-08-04
> > >
> > > Thank you for your response. We are glad that most of your concerns have been addressed, and we will incorporate these changes in the next version.
> > >
> > > On the question of the eigenvectors/eigenvalues of $J$ and $J^{-1}$, you are correct; a matrix and its inverse will share the same set of eigenvectors but with their eigenvalues flipped, i.e. for every eigenvector with $\lambda_i$ you would get the same eigenvector from the other matrix with eigenvalue $1/\lambda_i$.
> > >
> > > Here, showing that the top-K eigenvectors are similar for the two matrices could imply that the eigenvalues are tightly clustered around 1. This may help explain the fast convergence of our algorithm under the Inexact Newton lens, for which we know that as long as the spectrum of the Jacobian $J$ is well-bounded, we can perform $Je$ steps with a relatively large learning rate.

---

### Official Review · Reviewer_uiuT · 2025-07-03

**Clarity:** 2
**Significance:** 3
**Originality:** 3
**Rating:** 4
**Confidence:** 2

**Summary:**

This paper presents a novel algorithm to enhance the efficiency and quality of constrained image generation using pre-trained diffusion models. The authors identify a key trade-off in previous works, methods utilizing backpropagation are often slow and memory-intensive, while faster approaches that avoid it can fail to capture long-range dependencies. To address this issue, the proposed method approximates a Newton optimization step to apply constraints, avoiding computationally expensive backpropagation. The paper validates this approach on various linear  and non-linear tasks , demonstrating competitive performance in most cases.

**Questions:**

N/A

**Ethical Concerns:**

["NO or VERY MINOR ethics concerns only"]

**Final Justification:**

The authors have addressed my concerns and I remain my score of Borderline accept.

**Limitations:**

See weaknesses

**Quality:**

3

**Strengths And Weaknesses:**

**Strengths**

- The primary strength of this work lies in its computational efficiency. By approximating a Newton step using two forward passes instead of backpropagation , the algorithm effectively reduces inference time and memory requirements, making the approach highly practical for real-world applications.

- Despite its speed, the method produces high-quality results that are competitive with, and in most cases superior to, state-of-the-art training-free methods.

**Weaknesses**

- While the results are strong overall, the method's quantitative performance is not uniformly state-of-the-art. For example, in the 8x super-resolution task (Table 1), the reported FID score for "Ours" (60.99) is noticeably worse than that of the P2L baseline (51.81). An analysis of why the proposed method is less effective in this specific scenario would strengthen the paper.

- I noticed minor artifacts in some of the qualitative results. Specifically, in the second row of Figure 3 , the generated image of ours appears to have a glyph watermark. Could the authors explain this phenomenon? An explanation of the potential cause (e.g., artifacts from the pre-trained model or the optimization process itself) and possible mitigation strategies would be beneficial.

---

> ### Author Rebuttal · Authors · 2025-07-30
>
> We would like to thank the reviewer for taking the time to read our paper and provide valuable feedback. We respond to each of the weaknesses mentioned below:
>
> **Quantitative performance is not uniformly state-of-the-art**:
>
> In Table 1, we showed that the P2L baseline performs better on x8 super-resolution compared to our method. However, we point out that:
> 1. P2L requires 30 minutes to run on a single image, severely limiting its usability. Our method only requires 1 minute.
> 2. P2L uses a VLM (PaLI) to caption the low-resolution image before running the image sampling algorithm. We believe that for super-resolution, the text becomes integral in synthesizing the correct textures in the given image, which we hypothesized was the main advantage of P2L as mentioned in lines 207-208. Since all other previous methods only utilized the unconditional model, we also resorted to unconditional diffusion sampling in the main paper.
>
> To offer a complete picture, we verified the hypothesis that the advantage of P2L comes from using VLM-generated text prompts. We ran our 8x super-resolution algorithm with captions generated from an open-source VLM (Qwen-2.5 [Qwen]). For each low-res image, we first captioned it with the VLM using the instruction "Describe this image" to obtain a text prompt, and then applied our algorithm, conditioning the diffusion model on the VLM-generated prompt.
>
> The results presented below show that we improve upon the super-resolution PSNR and FID metrics of P2L when we use text captions with our proposed algorithm. When the text is correct, it accurately guides the generation of high-resolution details in the image. We observed that most current VLMs (GPT-4o, Qwen-2.5) are fully capable of providing meaningful captions even with low-resolution images.
>
> | Method | PSNR ↑ | LPIPS ↓ | FID ↓ |
> | - | - | - | - |
> | P2L (with VLM captions) | 23.38 | 0.386 | 51.81 |
> | Ours | 24.26 | 0.471 | 60.99 |
> | Ours + VLM captions | 24.95 | 0.405 | 44.74 |
>
> When it comes to comparing the various training-free sampling algorithms for super-resolution, adding a VLM to the pipeline introduces uncertainty, as the evaluations also need to consider the quality of the VLM and the specific prompt(s) used. Thus, we will include both results (super-res with and without text prompts) in the main paper and Table 1 for completeness.
>
> **Artifacts in generated images**:
>
> The artifact in Figure 3 row 2, is a watermark hallucinated by the model. This is solely an artifact of the image prior of Stable Diffusion, which was trained on (often watermarked) web images. We decided to use this image for Figure 3 because we found it interesting that our algorithm not only follows the conditions closely but also hallucinates novel high-frequency details under the constraint.
>
> Here, the hallucinated watermark is adhering to the imposed constraint. The mean squared error between the downsampled watermark image and the constraint is around 0.0008, which is also the mean squared error we get when we compare the constraint to another generated lighthouse image that has no hallucinated watermark. In contrast, previous methods often produce blurry results that lack hallucinated high-frequency textures, which the image prior should induce in the generated samples. For instance, all other methods in Figure 3 row 2 produce blurry textures for the rocks.
>
> To avoid confusion, we will replace this image with another sample that has no watermark hallucination, and move this example along with the discussion to Appendix **B.7 More Qualitative Results**.
>
> [Qwen] Bai, Shuai, et al. "Qwen2. 5-VL technical report." arXiv preprint 2025

---

### Official Review · Reviewer_WTsj · 2025-07-04

**Clarity:** 3
**Significance:** 2
**Originality:** 3
**Rating:** 3
**Confidence:** 4

**Summary:**

The paper introduces a fast, training-free algorithm for constrained sampling in pre-trained diffusion models, enabling high-quality image generation under arbitrary linear and non-linear constraints. By approximating Newton optimization steps using only forward passes through the denoiser, the method avoids expensive backpropagation while achieving results comparable to or better than state-of-the-art approaches. The technique is demonstrated on tasks like inpainting, super-resolution, and style-guided generation, significantly reducing inference time and memory usage.

**Questions:**

NA

**Ethical Concerns:**

["NO or VERY MINOR ethics concerns only"]

**Final Justification:**

After the discussion, I hold my opinion of borderline reject and open to any final decision

**Quality:**

2

**Strengths And Weaknesses:**

Strength: The  method proposes a new perspective for diffusion inverse problem to utilize the constraint signal besides backward gradient calculations.

Weakness:  My major concern is that the proposed method sacrifices performance to gain computation efficiency, it will be good to see the quantitive comparison of using newtown approximation method and using exact gradient descent, not just showing theoretical error analysis or visual comparison. Also SD1.4 has been three years old, would be good to use more recent backbones.

---

> ### Author Rebuttal · Authors · 2025-07-30
>
> We want to thank the reviewer for their comments and address their concerns below:
>
> **Sacrificing performance for efficiency**:
>
> We would like to point out that there is **no significant drop in performance** when using our method, while it requires just a fraction of the time to run. In Tables 2 and 3, we show that our algorithm is consistently superior to existing training-free approaches, especially in the more challenging box inpainting task, where it is only outperformed by the fine-tuned SD-Inpaint model, which required an additional 100k iterations of training.
>
> Regarding Table 1, although our algorithm performs worse than the P2L baseline in 8x super-resolution, P2L requires 30 minutes to run on a single image (vs 1min for ours) and also uses the PaLI VLM to caption the low-resolution images before the diffusion inference. As briefly mentioned in lines 207-208, we believe that the main advantage of P2L comes from introducing text conditioning, whereas all previous methods relied on the unconditional model. To verify this assumption, we ran the x8 super-resolution with text prompts generated from the downsampled images using Qwen-2.5 [Qwen] as the VLM. The results we present below show that we can indeed bridge the performance gap and even improve upon the PSNR and FID metrics when introducing text captions to the super-resolution process. We will add this result and relevant discussion to the paper to complement Table 1.
>
> | Method | PSNR ↑ | LPIPS ↓ | FID ↓ |
> | - | - | - | - |
> | P2L (with VLM captions) | 23.38 | 0.386 | 51.81 |
> | Ours | 24.26 | 0.471 | 60.99 |
> | Ours + VLM captions | 24.95 | 0.405 | 44.74 |
>
> To summarize, our method is both substantially faster and performs as well as or better than the baselines. There is no notable trade-off between efficiency and performance.
>
> **Approximate Newton vs gradient descent**:
>
> The goal of training-free algorithms is to compute and apply a gradient during the diffusion sampling that 'pushes' the generated image towards some desired condition. The previous methods we compare to in the paper (P2L, LDPS, PSLD, FreeDoM, MPGD) all use gradient descent to compute the update direction. Gradient descent is expressed as $J^Te$, where $J = \nabla f(x)$ is the denoiser Jacobian and $e$ the constraint error.
>
> We propose an alternative way to compute the update direction, based on approximating the steps a Newton method would make. The *exact* Newton method would use $J^{-1}e$ as the update direction, but this requires computing the inverse of the Jacobian, which is infeasible for these large diffusion models. We show that we can adequately approximate the Newton step with smaller updates in the direction of $Je$. This has been applied before in numerical optimization, as discussed in *Inexact Newton Methods* [5].
>
> Therefore, the comparison between our approximation to Newton's method and the exact gradient descent is shown in the experiments of the main paper. The previous methods using gradient descent are slower and produce worse results than the proposed approximate Newton method.
>
> **Approximate Newton vs exact Newton**:
>
> If the reviewer was referring to the accuracy approximating Newton's step $J^{-1}e$ with multiple inexact steps $\lambda Je$, unfortunately, there is no practical way to quantify it, as computing $J^{-1}$ for models at the scale of Stable Diffusion is prohibitive. In Equations (10)-(13), we discussed why our approximation converges to the correct result, but we are unable to directly compare the two update directions for the tasks shown in the paper.
>
> Regardless, we ran a smaller-scale experiment on MNIST where we can easily compute $J$ and $J^{-1}$ and show that both exhibit very similar structures. This aligns with the intuition we provided in lines 133-134 for using the proposed approximation. In the interest of space, we refer the reviewer to our response to Reviewer NFw9 for more details on this additional experiment.
>
> **Finite-difference approximation vs exact gradient**:
>
> To avoid any confusion regarding the term **approximation**, it's important to note here that to compute the proposed direction $Je$, we employ finite differences for the Jacobian-vector product $Je \approx (f(x + \delta e) - f(x))/\delta$. This is also an approximation, but for the Jacobian-vector product computation and **not** the constraint gradient update direction.
>
> The baselines that utilize gradient descent use automatic differentiation/backpropagation through the network. Our update can also be computed using automatic differentiation (forward-mode automatic differentiation). However, forward differentiation is not always implemented or optimized as well as the backward propagation of gradients.
>
> For completeness, we ran an ablation to study the choice of $\delta$ for the finite difference approximation and compare it to the exact gradient computation. For the forward gradient computation, we used the double backward trick since the forward gradient is not implemented for all Stable Diffusion layers in PyTorch. We ran the box inpainting task of Table 2 and found that our method (i) is robust to the choice of $\delta$, only breaking when using very small (0.0005) and very large (0.5) values, and (ii) performs as well as the exact Jacobian-vector product computation while requiring 1/5th of the inference time.
>
> | Method | PSNR ↑ | LPIPS ↓ | FID ↓ | Time |
> | - | - | - | - | - |
> Ours ($K=5$, $\lambda = 0.5$, $\delta = 0.0005$ ) | 17.47 | 0.37 | 69.02 | 15s |
> Ours ($K=5$, $\lambda = 0.5$, $\delta = 0.005$ ) | 18.30 | 0.30 | 42.01 | 15s |
> Ours ($K=5$, $\lambda = 0.5$, $\delta = 0.05$ ) | 18.32 | 0.31 | 42.44 | 15s |
> Ours ($K=5$, $\lambda = 0.5$, $\delta = 0.5$ ) | 15.80 | 0.34 | 61.40 | 15s |
> Ours ($K=5$, $\lambda = 0.5$, exact forward diff ) | 18.27 | 0.31 | 44.90 | 76s |
>
>  $K$ number of steps, $\lambda$ learning rate, $\delta$ finite difference step size
>
> Thus, using finite differences to compute the Jacobian-vector product is much faster and does not incur a drop in performance.
>
> **Using Stable Diffusion 1.4**:
>
> Stable Diffusion 1.4 has been widely used to benchmark training-free inference algorithms. Therefore, we also chose this model to showcase the advantages of our proposed method against the existing baselines. The proposed algorithm is not constrained to Stable Diffusion 1.4; we showcase qualitative results on a rectified flow model in the Appendix and provide an implementation for Stable Diffusion 2.1 in the supplementary code -- the results we get with SD-2.1 are similar to SD-1.4, but we did not perform a full-scale evaluation since all the baselines are evaluated only on SD-1.4.
>
> [Qwen] Bai, Shuai, et al. "Qwen2. 5-VL technical report." arXiv preprint 2025

---

> > ### Author Response · Authors · 2025-08-05
> >
> > Thank you once again for your time and comments. Given that the Author-Reviewer discussion period is coming to an end (Aug 8), we would like to ask if there are any additional concerns the reviewer may have after our rebuttal.

---

> > ### Comment · Reviewer_WTsj · 2025-08-08
> >
> > Dear Author,
> >
> > Thank you for your clarification on these points. My main concern is still the point that newtown approximation performs better than gradient descent both in regards of efficiency and performance. If this is the case, why machine learning researchers do not choose newtown approximation to optimize their neural network but with gradient descent. What makes this constrained sampling setting especially suitable for applying newtown approximation method considering the loss function is || Ax0-y||^2, with both better performance and efficiency. My impression is that newtown methods usually work on smooth, well-behaved, and small optimization problems like convex and finish much faster (quadratic convergence) than GD, but the problem here is relatively deep.
> >
> > Also another minor problem from me is that how do you deal with the newtown approximation when the Hessian is "unstable"  like singular or poorly conditioned which may result in large approximation error on these special samples, will this problem occur in this guided sampling scenario?
> >
> > Best,
> > WTsj

---

> > > ### Author Response · Authors · 2025-08-08
> > >
> > > Thank you for your response.
> > >
> > > We clarify that the results we presented in the paper are **specific to denoising diffusion models**.
> > >
> > > **Why Inexact Newton in denoising diffusion**:
> > >
> > > We believe that the relationship between $J$ and $J^{-1}$ in denoising diffusion makes the Inexact Newton method we propose suitable. That relationship is better understood with an intuitive example:
> > >
> > > Increasing a pixel's intensity in the noisy image should also increase the intensity in a neighborhood of that pixel in the predicted 'clean' image. We argue that, for denoising diffusion, the inverse should also be true: increasing the intensity of an output pixel should increase the intensity of all input noisy pixels in a local neighborhood. In both cases, the neighborhood size is determined by the timestep $t$. If there is a lot of noise, then that increase could be from far away pixels in the image, whereas for a small amount of noise, it's most likely that only that one pixel value was increased.
> > >
> > > If we formulate this in terms of the learned denoiser $\hat{x}_0(x_t)$ and its Jacobian $J$, we can probe how a change $e$ in the output should be applied to the input using the learned input $\rightarrow$ output transformation, i.e. $Je$, instead of first inverting the transformation, which involves computing $J^{-1}e$.
> > >
> > > We also performed a smaller-scale experiment in our response to Reviewer NFw9, where we computed and compared the structures of $J$ and $J^{-1}$ in denoising diffusion models. We refer the reviewer to that to further motivate our proposed Inexact Newton step.
> > >
> > > **Convexity of the constraint**:
> > >
> > > The applicability of our method to denoising diffusion models is not related to the convexity of the constraint we want to apply. As you correctly pointed out, the tough part is propagating the error through the denoiser, which is hundreds of millions of parameters deep. Our method works because of the structure of the diffusion model itself, as we discussed above.
> > >
> > > **Why not Inexact Newton everywhere**:
> > >
> > > For optimizing neural network parameters in general, the Inexact Newton method we present (using $Je$ instead of $J^{-1}e$) is not applicable when J is not square, which is most (or all) neural network optimization cases. Moreover, it is probably not the optimal choice when the input-output transformation modeled by $J$ and $J^{-1}$ (how changes between parameters and network outputs are related) is more complex than denoising, which has the nice properties we discussed above.
> > >
> > > **Improvement over gradient descent**:
> > >
> > > On efficiency, the Inexact Newton method is better because of the finite difference approximation of the Jacobian-vector product $Je$. This finite-difference approximation turns out to be faster and less memory demanding than backpropagating through the model, which is required for gradient descent.
> > >
> > > On performance, for an ideal denoiser, the Jacobian should be symmetric (Eq .(17)), making the Inexact Newton method equivalent to gradient descent. In practice, we find that the Jacobian is not exactly symmetric (Appendix A.2), giving us different results from gradient descent. This non-symmetry of the Jacobian could be one of the advantages of our method, but it is difficult to exactly measure for Stable Diffusion-scale models, where computing Jacobians is prohibitive. Empirically, we find that the steps produced by our method capture better long-range correlations between pixels (Figure 2), and that we can use a larger learning rate without diverging.
> > >
> > > **Stability of the Hessian**:
> > >
> > > We use an Inexact Newton method to solve the nonlinear system $Je_t = e$ for $e_t$. We get this system of equations using **only the first-order terms** from the Taylor approximation, as shown in Eq. (5). Since no second-order terms are involved, the conditioning of the Hessian does not affect our method differently from how it would affect gradient descent (also first-order only).
> > >
> > > We understand that there may be some confusion around the term "Newton method", since it is widely used to refer to optimization algorithms that use second-order terms (the Hessian or an approximation of it). In our case, we only use the first-order Jacobian matrix.

---

### Decision · Program_Chairs · 2025-09-17

**Decision:**

Accept (poster)

**Comment:**

This paper proposes an algorithm for performing diffusion sampling under target constraints. It's often time-consuming to finetune the pretrained diffusion model on the new constraints and the cost can be reduced through constrained sampling. The authors propose to use second-order method under linear constraints to perform sampling for a diffusion model and approximate the jacobian inverse with jacobian. The proposed algorithm runs much more efficient compared to baselines and achieves on-par results. Reviewers are concerned that the performance is sacrificed (reviewer WTsj), not state-of-the-art (reviewer uiuT), the approximation of jacobian (reviewer, NFw9), and lack of novelties. The discussion has cleared most of the concerns among reviewers.  Reviewer WTsj is concerned on trading performance with efficiency. The algorithm demonstrates a strong gain on efficiency while the performance fluctuation is within some reasonable range. I'd suggest this paper as a candidate for acceptance.